



# Suitability of 17 rainfall and temperature gridded datasets for large-scale hydrological modelling in West Africa

Moctar Dembélé[1], Bettina Schaefli[1,3], Nick van de Giesen[2] & Grégoire Mariéthoz[1]

[1]Institute of Earth Surface Dynamics, Faculty of Geosciences and Environment, University of Lausanne, CH-1015 Lausanne, Switzerland
[2]Water Resources Section, Faculty of Civil Engineering and Geosciences, Delft University of Technology, Stevinweg 1, 2628 CN Delft, The Netherlands
[3]Now at: Institute of Geography, Faculty of Science, University of Bern, CH-3012, Switzerland

*Correspondence to*: Moctar Dembélé (moctar.dembele@unil.ch)

**Abstract.** This study evaluates the ability of different gridded rainfall datasets to plausibly represent the spatiotemporal patterns of multiple hydrological processes (i.e. streamflow, actual evaporation, soil moisture and terrestrial water storage) for large-scale hydrological modelling in the predominantly semi-arid Volta River Basin (VRB) in West Africa. Seventeen precipitation products based on satellite data (TAMSAT, CHIRPS, ARC, RFE, MSWEP, GSMaP, PERSIANN-CDR, CMORPH-CRT, TRMM 3B42, TRMM 3B42RT) and on reanalysis (JRA-55, EWEMBI, WFDEI-GPCC, WFDEI-CRU, MERRA-2, PGF and ERA5) are compared as input for the fully distributed mesoscale Hydrologic Model (mHM). To assess the model sensitivity to meteorological forcing during rainfall partitioning into evaporation and runoff, six different temperature reanalysis datasets are used in combination with the precipitation datasets, which results in evaluating 102 combinations of rainfall-temperature input data. The model is recalibrated for each of the 102 input combinations, and the model responses are evaluated by using in-situ streamflow data and satellite remote sensing datasets from GLEAM evaporation, ESA CCI soil moisture, and GRACE terrestrial water storage. A bias-insensitive metric is used to assess the impact of meteorological forcing on the simulation of the spatial patterns of hydrological processes. The results of the process-based evaluation show that the rainfall datasets have contrasting performances across the four climatic zones present in the VRB, suggesting that, in general, basin-wide hydrological model performance might be misleading and invalid for a smaller spatial domain. No single rainfall or temperature dataset consistently ranks first in reproducing the spatiotemporal variability of all hydrological processes. A dataset that is best in reproducing the temporal dynamics is not necessarily the best for the spatial patterns. In addition, the results suggest that there is more uncertainty in representing the spatial patterns of hydrological processes than their temporal dynamics. Finally, some region-tailored datasets outperform the global datasets, thereby stressing the necessity and importance of regional evaluation studies for satellite and reanalysis meteorological datasets.

**Keywords:** Precipitation; Atmospheric forcing; Hydrological consistency; Process-based evaluation; Data uncertainty propagation; Ungauged basins; Data scarce regions



## 1 Introduction

Our understanding of environmental systems is underpinned by observational data whose unavailability and uncertainties hinder research and operational applications. Among other factors, atmospheric data quality is of prime importance for the

reliability of hydro-meteorological and climatological studies (Ledesma and Futter, 2017;Zandler et al., 2019). Precipitation is one of the major components of the water cycle, which has led to numerous initiatives on understanding its generation, and estimating its amount and variability on Earth (Maidment et al., 2015;Cui et al., 2019). In hydrological modelling (Singh, 2018;Beven, 2019), precipitation is the most important driver variable that determines the spatiotemporal variability of other hydrological fluxes and state variables (Thiemig et al., 2013;Bárdossy and Das, 2008).

With the development of distributed hydrological models that facilitate large-scale predictions (Clark et al., 2017;Fatichi et al., 2016;Ocio et al., 2019), there is a growing need to inform and evaluate those models with distributed observational datasets to improve spatiotemporal process representation (Baroni et al., 2019;Paniconi and Putti, 2015;Hrachowitz and Clark, 2017). A key challenge is the spatiotemporal intermittency of precipitation, which is a major challenge for its measurement and its spatial interpolation (Tauro et al., 2018;Acharya et al., 2019;Bárdossy and Pegram, 2013;Wagner et al., 2012a) , especially in

regions with particular features such as complex topography, convection-driven precipitation or snowfall occurrence. A comprehensive description of precipitation measurement techniques can be found in previous studies (e.g. Tapiador et al., 2012;Stephens and Kummerow, 2007;Kidd and Huffman, 2011). The drawbacks of in-situ measurements of precipitation include limited and uneven areal coverage, deficiencies in instruments and costly maintenance (Kidd et al., 2017;Awange et al., 2019;Harrison et al., 2019), and have led to the advent of precipitation estimation from space (Barret and Martin, 1981).

Precipitation estimates from space are spatially homogeneous and cover inaccessible regions with uninterrupted records over time (Beck et al., 2019b;Funk et al., 2015).

The advent of satellite-based rainfall products (SRPs) has opened up new avenues for water resources monitoring and prediction, especially in data sparse regions (Serrat-Capdevila et al., 2014;Sheffield et al., 2018;Hrachowitz et al., 2013). Although, the use of SRPs in hydrology is increasing (Xu et al., 2014;Chen and Wang, 2018), they have not been fully adopted

for operational purposes yet (Ciabatta et al., 2016;Kidd and Levizzani, 2011). The limited uptake of SRPs in hydrology is due to measurement bias, inadequate spatiotemporal resolutions (e.g. for extreme event simulation) and shortness of the records for some applications (e.g., climate change impact assessments), and the skepticism of some potential users with regard to the data quality (Marra et al., 2019). In the past decades, a large number of SRPs have been developed with different objectives, spatial and temporal resolutions, input sources, algorithms and acquisition methods (Ciabatta et al., 2018;Ashouri et al.,

2015;Brocca et al., 2019). Several studies provide a review of SRPs (e.g. Maidment et al., 2014;Sun et al., 2018;Maggioni et al., 2016;Le Coz and van de Giesen, 2019).

In addition to SRPs, there are also atmospheric retrospective analysis (or reanalysis) datasets of precipitation. A reanalysis system is composed of a forecast model and a data assimilation scheme that integrates spatiotemporal observations of meteorological variables (i.e. temperature, humidity, wind and pressure) to generate gridded atmospheric data (Lorenz and



65 Kunstmann, 2012;Schröder et al., 2018). Precipitation is one of the reanalysis model-generated fields that generally has more uncertainties than the meteorological state fields (Roca et al., 2019). Reanalysis datasets are often used in hydrological modelling (Tang et al., 2019;Duan et al., 2019;Gründemann et al., 2018), and sometimes they are preferred over SRPs because of their usually long-term records suitable for climate change studies, and because of their higher performance in predictable large-scale stratiform systems (Seyyedi et al., 2015;Potter et al., 2018).

70 Despite the progress in satellite instruments, which has led to substantial advances in improving precipitation estimates (Sorooshian et al., 2011;Tang et al., 2019), there are known inconsistencies among the available SRPs (Sun et al., 2018;Tapiador et al., 2017). SRPs are subject to inherent errors originating mainly from precipitation retrieval instruments and algorithms, sampling frequency, and inadequate representation of cloud physics in some regions (Laiti et al., 2018;Alazzy et al., 2017;Romilly and Gebremichael, 2011). While on the one hand SRPs are subject to systematic biases, reanalysis products

75 on the other hand have uncertainties resulting from their model forcing parameters, low spatial resolution with poor representation of sub-grid processes, and the model physics (Bosilovich et al., 2008;Laiti et al., 2018). Uncertainty quantification both in SRPs and reanalysis data is subject to intense research (e.g. Maggioni et al., 2016;Gebremichael, 2010;Awange et al., 2016;Westerberg and Birkel, 2015). The errors quantification of SRPs and reanalysis products is usually done by comparing them with in-situ measurements (e.g. Dembélé and Zwart, 2016;Thiemig et al., 2012;Beck et al.,

80 2019a;Caroletti et al., 2019;Satgé et al., 2020), or by assessing their reliability as forcing for hydrological models (e.g.Duethmann et al., 2013;Pan et al., 2010;Nkiaka et al., 2017). Other evaluation approaches include triple collocation, which is a technique that estimates the variance of unknown errors of three independent variables without a reference or observed variable (e.g. Massari et al., 2017;Alemohammad et al., 2015;McColl et al., 2014;Roebeling et al., 2012). Compared to the ground-truthing approach, the hydrological evaluation approach has received limited attention (Camici et al., 2018;Poméon et

85 al., 2017).

In rainfall-runoff modelling (Wagener et al., 2004;Beven, 2011), the non-linearity of hydrological processes (Blöschl and Zehe, 2005;Clark et al., 2009) can reduce or amplify the errors in the used input rainfall data and result in a satisfactory or poor representation of the hydrological responses (Maggioni and Massari, 2018;Nijssen, 2004). Consequently, the hydrological model can give a good representation of a hydrological state or flux variable for the wrong reasons (cf. Kirchner, 2006), thereby

90 potentially leading to unfortunate consequences for water resources management (Zambrano-Bigiarini et al., 2017). When testing models as hypotheses (Beven, 2018;Pfister and Kirchner, 2017), type I errors (i.e. false positive model acceptability; Beven, 2010) should be avoided to ensure a high predictive skill of the model and its correctness for good decision-making. This sheds light on the importance of assessing the reliability of hydrological predictions generated with the use of SRPs and reanalysis products (Behrangi et al., 2011;Kuczera et al., 2010). In this context, knowing the adequacy and coherence of

95 meteorological data in reproducing hydrological processes is a prerequisite to data selection for water resources management (Casse et al., 2015;Laiti et al., 2018).

In the context of hydrological evaluation of precipitation datasets, some limitations can be identified in previous studies. Some studies only evaluate a small number of precipitation datasets or do not consider reanalysis products (e.g. Bitew and





Gebremichael, 2011;Ma et al., 2018;Liu et al., 2017;Bhattacharya et al., 2019). Usually, the influence of temperature datasets
in combination with rainfall datasets is not tested (e.g. Satgé et al., 2019;Camici et al., 2018;Casse et al., 2015;Qi et al.,
2016;Zhang et al., 2019), with the exception of a few studies (e.g. Laiti et al., 2018;Lauri et al., 2014), despite the importance
of this interaction for evaporation simulation. Most studies evaluate a single hydrological state or flux variable, generally
streamflow (e.g. Poméon et al., 2017;Seyyedi et al., 2015;Shayeghi et al., 2020;Li et al., 2012b), or soil moisture (e.g. Brocca
et al., 2013). Some studies use lumped or semi-distributed models, therefore averaging the rainfall amount on large areas (e.g.
Duan et al., 2019;Tang et al., 2019;Tobin and Bennett, 2014;Gosset et al., 2013;Shawul and Chakma, 2020), which reduces
the bias effect that could occur at the pixel level with a fully distributed model. Often, the model is not recalibrated for each
precipitation dataset (e.g. Voisin et al., 2008;Su et al., 2008;Li et al., 2012a;Tramblay et al., 2016), which is, however, a
prerequisite for reliable input field assessment (Stisen et al., 2012). Moreover, some studies perform a global-scale analysis
and ignore regionally tailored products (e.g. Beck et al., 2017b;Mazzoleni et al., 2019;Fekete et al., 2004), which can
outperform global products (e.g. Thiemig et al., 2013). Finally, to the best of our knowledge, no study evaluated the
simultaneous impact of various precipitation and temperature datasets on the spatial patterns of several hydrological processes
(i.e. soil moisture and evaporation).

In light of the above, we propose to study the adequacy of different combinations of 17 precipitation datasets (10 SRPs and 7
reanalysis products) and 6 temperature datasets from reanalysis, when used as forcing data for a fully distributed hydrological
model, in reproducing the spatiotemporal variability of multiple hydrological processes (i.e. streamflow, actual evaporation,
soil moisture, and terrestrial water storage). In total, 102 rainfall-temperature input data combinations are tested with the
mesoscale Hydrologic Model (mHM) by recalibrating the model for each of the input data combinations. The experiment is
carried out in the poorly gauged and predominantly semi-arid Volta River Basin (VRB) located in West Africa, over the period
2003-2012. It is noteworthy that the goal of this study is not to estimate the intrinsic quality of the meteorological forcing (i.e.
precipitation and temperature) but rather to understand the impact of the propagation of associated uncertainties on the
simulation of hydrological processes (Bhuiyan et al., 2019;Falck et al., 2015;Marthews et al., 2020).

The VRB case study is particularly interesting from both scientific and societal perspectives. On the one hand, precipitation
modelling in tropical monsoon climates is a challenging task due to strong seasonality and diurnal variations of rainfall (Turner
et al., 2011;Pfeifroth et al., 2016;Cook and Vizy, 2019), and due to isolated convection systems in semi-arid regions (Taylor
et al., 2017;Mathon et al., 2002;Parker and Diop-Kane, 2017). On the other hand, open access and good quality datasets are
needed for water resources management in West Africa (Roudier et al., 2014;Serdeczny et al., 2017;Di Baldassarre et al.,
2010;Dinku, 2019). The following research questions are addressed:

    1) What is the impact of different gridded rainfall and temperature datasets on the simulation of hydrological fluxes and
       state variables?

2) How important is the choice of meteorological datasets for the representation of spatial patterns versus temporal
       dynamics?





Overall, the objective of this work aligns with the efforts in solving the current scientific challenges in hydrology (i.e. uncertainty in large-scale measurements and data, spatial heterogeneity and modelling methods; Blöschl et al., 2019;Wilby, 2019). Moreover, a growing interest in using satellite remote sensing data in hydrological modelling is expected (McCabe et

al., 2017;Peters-Lidard et al., 2017;Wilkinson et al., 2016). Therefore, knowing the suitability of the input data for hydrological modelling is a prerequisite for reliable spatiotemporal predictions, as the goal is to increase model performance with minimum uncertainty (Beven, 2016;McMillan et al., 2018;Savenije, 2009).

## 2 Methodology

### 2.1 Overview of the modelling experiment

The adequacy of the rainfall and temperature datasets to plausibly reproduce various hydrological processes is tested with all the 102 possible combinations of 17 rainfall and 6 temperature datasets used as meteorological forcing (see section 2.2). Different temperature datasets are used to allow flexibility in rainfall partitioning into evaporation and runoff because temperature is a key variable for the calculation of potential evaporation (Kirchner and Allen, 2020;Zheng et al., 2019;Van Stan et al., 2020). The hydrological model is recalibrated for each of the 102 combinations of rainfall-temperature datasets

(Figure 1).

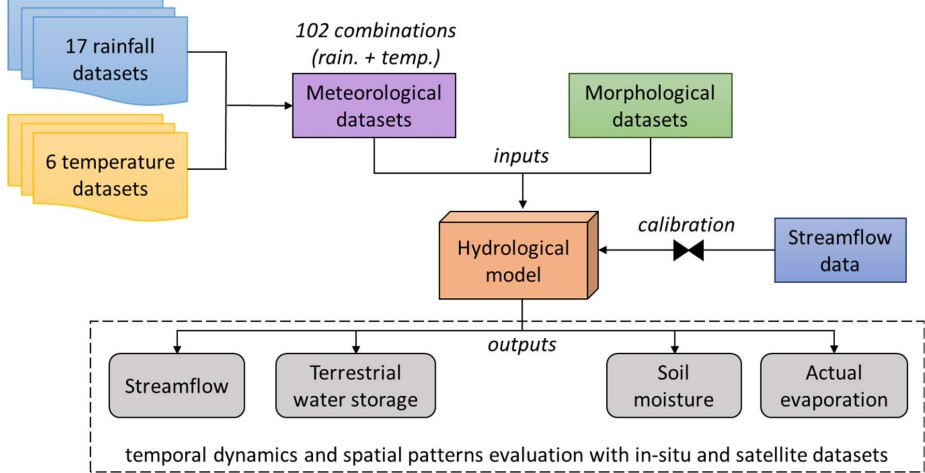

**Figure 1. Flowchart of the methodology used to evaluate the suitability of meteorological datasets in reproducing plausible hydrological processes.**

The differences in the performance of model outputs are assumed to result from the propagation of the input data uncertainty

through the model simulations (Nikolopoulos et al., 2010;Fallah and Orth, 2020). In case of uncertainties resulting from the hydrological model structure, these uncertainties can be assumed to remain consistent for all the input datasets and therefore



it should not hinder the interpretation of the results, because only the parameters change during model calibration and not the model structure (Raimonet et al., 2017).

### 2.2 Meteorological datasets

This study evaluates 17 rainfall products composed of 10 satellite-based products: TAMSAT, CHIRPS, ARC, RFE, MSWEP, GSMaP, PERSIANN-CDR, CMORPH-CRT, TRMM 3B42 and TRMM 3B42RT; and 7 reanalysis products: JRA-55, EWEMBI, WFDEI-GPCC, WFDEI-CRU, MERRA-2, PGF and ERA5 (Table 1). Widely used global and Africa-tailored datasets were selected based on their availability in the period for which streamflow data is available for the hydrological modelling (2000-2012). For SRPs having multiple versions, the gauge-corrected version was selected to avoid the known systematic biases found in the SRPs as compared to ground measurements (Jiang and Wang, 2019;Pellarin et al., 2020). The selected rainfall datasets include single and multi-sensor, with various merged and gauge-corrected products obtained from rain gauges, microwave sensors on low Earth orbits and infrared sensors on geostationary satellites (Maggioni and Massari, 2018;Thiemig et al., 2013;Golian et al., 2019). Moreover, six different datasets of air temperature (at 2 m above ground) are used for the calculation of potential evaporation and they are obtained from the reanalysis products: JRA-55, EWEMBI, WFDEI, MERRA-2, PGF and ERA5.

**Table 1.** Meteorological datasets with used spatial resolution; the table presents the characteristics of the datasets used in this study, although different spatial and temporal resolutions can be available from the data providers. G: gauge, S: satellite, R: reanalysis, NP: near-present.

| Datasets | Name/ website | Data sources | Spatial coverage | Spatial resolution | Temporal coverage | Temporal resolution | References |
|---|---|---|---|---|---|---|---|
| **TAMSAT v3.0** | Tropical Applications of Meteorology using SATellite (TAMSAT), African Rainfall Climatology and Time-series (TARCAT) https://www.tamsat.org.uk/data/archive | S, G | Africa 38°N – 36°S, 19°W – 52°E | 0.0375° | 1983-NP | daily | Maidment et al. (2017), Tarnavsky et al. (2014), Maidment et al. (2014) |
| **CHIRPS v2.0** | Climate Hazards group InfraRed Precipitation with Stations (CHIRPS) V2.0 http://chg.ucsb.edu/data/chirps/ | S, G, R | Land 50° N/S, 180° E/W | 0.05° | 1981-NP | daily | Funk et al. (2015) |
| **ARC v2.0** | Africa Rainfall Estimate Climatology (ARC 2.0) https://www.cpc.ncep.noaa.gov/products/international/data.shtml | S, G | Africa 40°N – 40°S, 20°W – 55°E | 0.1° | 1983-NP | daily | Novella and Thiaw (2013) |
| **RFE v2.0** | Climate Prediction Center (CPC) African Rainfall Estimate (RFE) https://www.cpc.ncep.noaa.gov/products/international/data.shtml | S, G | Africa 40°N – 40°S, 20°W – 55°E | 0.1° | 2001-NP | daily | Xie and Arkin (1996), Herman et al. (1997) |



| | | | | | | | |
|---|---|---|---|---|---|---|---|
| **MSWEP v2.2** | Multi-Source Weighted-Ensemble Precipitation (MSWEP) V2.2 http://www.gloh2o.org/ | S, G, R | Global | 0.1° | 1979-NP | 3-hourly | Beck et al. (2017a) |
| **GSMaP-std v6** | Global Satellite Mapping of Precipitation (GSMaP) Moving Vector with Kalman (MVK) Standard V6 https://sharaku.eorc.jaxa.jp/GSMaP/ | R, G | 60° N/S, 180° E/W | 0.1° | 2001-2013 | daily | Ushio et al. (2009), Ushio et al. (2019) |
| **PERSIANN-CDR v1r1** | Precipitation Estimation from Remotely Sensed Information using Artificial Neural Networks (PERSIANN) Climate Data Record (CDR) V1R1 http://chrsdata.eng.uci.edu/ | S, G | 60° N/S, 180° E/W | 0.25° | 1983-2016 | 6-hourly (daily) | Ashouri et al. (2015) |
| **CMORPH-CRT v1.0** | Climate Prediction Center (CPC) MORPHing technique (CMORPH) bias corrected (CRT) V1.0 www.cpc.ncep.noaa.gov | S, G | 60° N/S, 180° E/W | 0.25° | 1998-2015 | daily | Joyce et al. (2004), Xie et al. (2017) |
| **TRMM 3B42 v7** | TRMM Multi-satellite Precipitation Analysis (TMPA) 3B42 V7 https://mirador.gsfc.nasa.gov/ | S, G | 50° N/S, 180° E/W | 0.25° | 2000-2017 | 3-hourly | Huffman et al. (2007) |
| **TRMM 3B42 RT v7** | TRMM Multi-satellite Precipitation Analysis (TMPA) 3B42 Real Time V7 https://mirador.gsfc.nasa.gov/ | S | 50° N/S, 180° E/W | 0.25° | 2000-NP | 3-hourly | Huffman et al. (2007) |
| **WFDEI-CRU** | WATCH Forcing Data ERA-Interim (WFDEI) corrected using Climatic Research Unit (CRU) dataset www.eu-watch.org | R, G | Global | 0.5° | 1979-2018 | 3-hourly | Weedon et al. (2014) |
| **WFDEI-GPCC** | WATCH Forcing Data ERA-Interim (WFDEI) corrected using Global Precipitation Climatology Centre (GPCC) dataset ftp://rfdata:forceDATA@ftp.iiasa.ac.at/ | R, G | Global | 0.5° | 1979-2016 | 3-hourly | Weedon et al. (2014) |
| **PGF v3** | Princeton University global meteorological forcing (PGF) http://hydrology.princeton.edu/data/pgf/ | R, G | Global | 0.25° | 1948-2012 | 3-hourly | Sheffield et al. (2006) |
| **ERA5** | European Centre for Medium-range Weather Forecasts ReAnalysis 5 (ERA5) hourly data on single levels https://cds.climate.copernicus.eu/ | R | Global | 0.25° | 1979-NP | hourly | Hersbach et al. (2018) |
| **MERRA-2** | Modern-Era Retrospective Analysis for Research and Applications 2 (rainfall: | S, G, R | Global | 0.625° x 0.5° | 1980-NP | hourly | Gelaro et al. (2017), |





| | M2T1NXFLX_V5.12.4; temperature: M2SDNXSLV_V5.12.4) https://disc.gsfc.nasa.gov/datasets/ | | | | | | Reichle et al. (2017) |
|---|---|---|---|---|---|---|---|
| **EWEMBI v1.1** | EartH2Observe, WFDEI and ERA-Interim data Merged and Bias-corrected for ISIMIP (EWEMBI) http://doi.org/10.5880/pik.2016.004 | R, G | Global | 0.5° | 1976-2013 | daily | Lange (2016) |
| **JRA-55** | Japanese 55 year ReAnalysis (JRA-55); rainfall: fcst_phy2m125; temperature: anl_surf125 https://jra.kishou.go.jp/JRA-55/index_en.html | R | Global | 1.25° | 1959-NP | 3-hourly | Kobayashi et al. (2015) |

### 2.3 Modelling datasets

In addition to the meteorological datasets (Table 1), an ensemble of datasets is required for the set-up and the calibration and

175   evaluation of the hydrological model (Table 2). The streamflow datasets obtained from different organizations (see acknowledgements) were pre-processed (i.e. gap-filling and quality control) in the work of Dembélé et al. (2019).

**Table 2. Modelling datasets. ESA CCI SM: European Space Agency Climate Change Initiative Soil Moisture; GIMMS: Global Inventory Modelling and Mapping Studies; GLEAM: Global Land Evaporation Amsterdam Model; GLiM: Global Lithological**
180   **Map; GMTED: Global Multi-resolution Terrain Elevation Data; GRACE: Gravity Recovery and Climate Experiment; WFDEI: WATCH Forcing Data methodology applied to ERA-Interim data.**

| Variables | Products | Spatial resolution | Temporal resolution | References |
|---|---|---|---|---|
| **Morphological data** | | | | |
| Terrain characteristics (elevation, slope, aspect, flow direction and flow accumulation) | GMTED 2010 | 225 m (0.0021°) | static | Danielson and Gesch (2011) https://topotools.cr.usgs.gov/ |
| Soil properties (horizon depth, bulk density, sand and clay content,) | SoilGrids | 250 m (0.0023°) | static | Hengl et al. (2017) https://www.isric.org/explore/soilgrids |
| Geology | GLiM v1.0 | 0.5° | static | Hartmann and Moosdorf (2012) https://doi.pangaea.de/10.1594/PANGAEA.788537 |
| Land use land cover | Globcover 2009 | 300 m (0.0028°) | static | Bontemps et al. (2011) http://due.esrin.esa.int/page_globcover.php |





| Phenology (leaf area index) | GIMMS | 8 km (0.0833°) | bimonthly | Tucker et al. (2005), Zhu et al. (2013) http://cliveg.bu.edu/modismisr/lai3g-fpar3g.html |
|---|---|---|---|---|
| **Model calibration/evaluation** | | | | |
| Streamflow | - | point | daily | Multiple organizations (see acknowledgements) |
| Terrestrial water storage anomaly ($S_t$) | GRACE TellUS v5.0 | 1° | monthly | Tapley et al. (2004), Landerer and Swenson (2012) https://grace.jpl.nasa.gov/ |
| Surface soil moisture ($S_u$) | ESA CCI SM v4.2 | 0.25° | daily | Dorigo et al. (2017) https://www.esa-soilmoisture-cci.org/ |
| Actual evaporation ($E_a$) | GLEAM v3.2a | 0.25° | daily | Martens et al. (2017), Miralles et al. (2011) https://www.gleam.eu/ |

Multiple satellite datasets are used to evaluate the modelled hydrological fluxes and state variables. For the evaluation of the modelled water storages, the GRACE-derived terrestrial water storage ($S_t$) anomaly data release RL05 (Landerer and Swenson, 2012;Swenson, 2012) is used. The ensemble mean of different products from three processing centers (i.e. Jet Propulsion Laboratory, Center for Space Research at University of Texas, and Geoforschungs Zentrum Potsdam) is preferred because it is more effective in reducing noise in the Earth's gravity signal as compared to the individual products (Sakumura et al., 2014). The surface soil moisture ($S_u$) data representing the first soil layer (i.e. 2-5 cm depth) is obtained from ESA CCI (Dorigo et al., 2017) using the combination of both active and passive microwave products (Gruber et al., 2017;Wagner et al., 2012b). Actual evaporation ($E_a$) data is obtained from the GLEAM land surface model that aggregates components of terrestrial evaporation based on the fraction of land cover types per grid cell (Martens et al., 2017). A full description of the datasets is accessible through the references and web links provided in Table 1 and Table 2.

### 2.4 Study Area

The transboundary Volta River Basin (VRB) covers approximately 415,600 km² (Figure 2) shared among six countries of West Africa (i.e. Burkina Faso, Ghana, Togo, Mali, Benin and Côte d'Ivoire). The relief is predominantly flat with 95% of the basin below 400 m a.s.l (De Condappa and Lemoalle, 2009). The Volta River flows over 1,850 km with a drainage system composed of four sub-basins known as Black Volta (152,800 km²), White Volta (113,400 km²), Oti (74,500 km²), and Lower Volta (74,900 km²). Before reaching the Atlantic Ocean at the Gulf of Guinea, the Volta River transits in the Lake Volta (area: 8,502 km²; volume: 148 km³) formed by the Akosombo dam (7.94 10⁶ m³) (Williams et al., 2016;Dembélé et al., 2020). The dominant land cover is savannah composed of grassland interspersed with shrubs and trees over 75% of the basin area, followed by cropland (13%), forest (9%), water bodies (2%) and bare land and settlements (1%). Climate in West Africa is unique and complex (Berthou et al., 2019;Bichet and Diedhiou, 2018;Nicholson et al., 2018a). The seasonal and latitudinal oscillation of the Inter-Tropical Convergence Zone (ITCZ) is the predominant rainfall generation mechanism in West Africa (Biasutti, 2019), thereby depicting a south-north gradient of increasing aridity in the VRB. The ITCZ is a narrow belt of clouds associated with





intense convective activity resulting from the near-surface convergence of warm and moist trade winds (Schneider et al., 2014;Dezfuli, 2017). The warm northeasterly Harmattan winds emanate from the Sahara and the moist southwest monsoon winds originate in the Atlantic ocean (Nicholson, 2013;Vizy and Cook, 2018). Rainfall in West Africa is characterized by its interannual and multidecadal variability (Biasutti et al., 2018;Thorncroft et al., 2011;Nicholson et al., 2018b). Four eco-
climatic zones (i.e. Sahelian, Sudano-Sahelian, Sudanian and Guinean; Table 3) are commonly identified based on the average annual precipitation and agricultural features (FAO/GIEWS, 1998;Mul et al., 2015). The aridity index in Table 3 is derived from the global aridity index database (Trabucco and Zomer, 2018). The maps of spatial patterns of rainfall and temperature in the VRB for different datasets are shown in Appendix A1 and Appendix A2. The climatology of rainfall and temperature per climatic zones are provided in the Supporting Information (SI, Figures S3-S6).

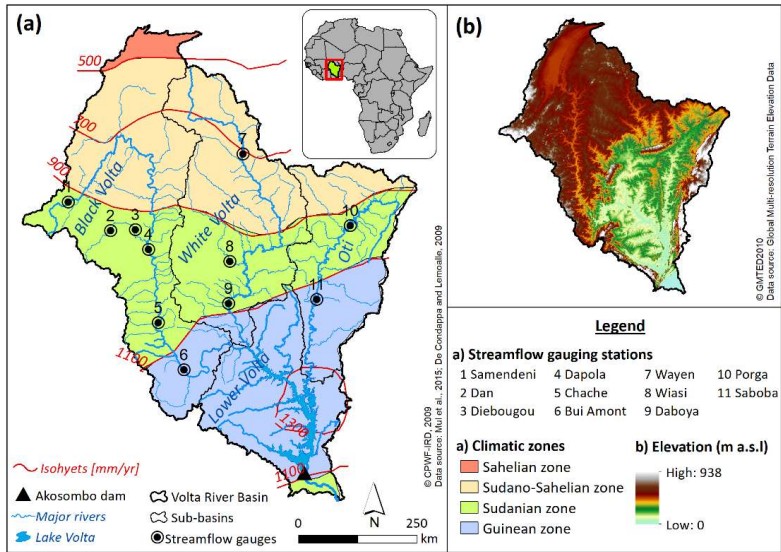


**Figure 2. Physical and hydroclimatic characteristics of the Volta River basin.**

**Table 3. Characteristics of the four eco-climatic zones in the Volta River basin. The mean and range ([min-max]) values are given for the Aridity Index (*AI*).**

| Eco-climatic zones | Climate class | *AI* (-) |
|---|---|---|
| Sahel Savanna | Arid | 0.16 [0.12-0.20] |
| Sudano-Sahelian | Semi-arid | 0.29 [0.16-0.43] |
| Sudanian Savanna | Semi-arid/ Dry sub-humid | 0.47 [0.33-0.98] |
| Guinean Savanna | Dry sub-humid/ Humid | 0.70 [0.49-1.22] |






### 2.5 Hydrological Model Setup

The fully distributed mesoscale Hydrologic Model (mHM, version 5.9; Samaniego et al., 2010;Kumar et al., 2013) is used in this study. It is a conceptual model that simulates dominant hydrological processes (e.g. evaporation, soil moisture, subsurface storage, and discharge) per grid cell in the modelling domain. The Muskingum-Cunge method (Cunge, 1969) is used for

routing the total grid-generated runoff using a multiscale routing model (Thober et al., 2019). A multiscale parameter regionalization technique (MPR; Samaniego et al., 2017) is used to account for sub-grid variability of the basin physical characteristics (e.g. soil texture, topography and land cover). For this study, 36 global parameters are determined through model calibration (Table S18 in the Supporting Information).

In this study, the Hargreaves and Samani method (Hargreaves and Samani, 1985), solely based on air temperature data, is used

to calculate the reference evaporation ($E_{ref}$). Potential evaporation ($E_p$) is calculated by adjusting $E_{ref}$ to vegetation cover (Allen et al., 1998;Birhanu et al., 2019). A dynamical scaling function ($F_{DS}$) (cf. Demirel et al., 2018) is used to account for vegetation-climate interactions (Bai et al., 2018;Jiao et al., 2017). $E_p$ is formulated as follows:

$$E_p = F_{DS} \cdot E_{ref}, \text{ with} \qquad\qquad (1)$$

$$F_{DS} = a + b\left(1 - e^{(c \cdot I_{LA})}\right) \qquad\qquad (2)$$

where $I_{LA}$ represents the leaf area index, $a$ is the intercept term, $b$ represents the vegetation dependent component, and $c$ describes the degree of nonlinearity in the $I_{LA}$ dependency. The coefficients $a$, $b$, and $c$ are determined during model calibration.

Actual evaporation (i.e. all evaporative fluxes including transpiration, $E_a$) depends on plant water availability, i.e. on  root distribution in the subsurface and soil moisture availability (Feddes et al., 1976); this is emulated in mHM by computing $E_a$ as a fraction of $E_p$ at different soil layers. A multi-layer infiltration capacity approach is used to calculate soil moisture based on a three-layer soil scheme (5 cm, 30 cm and 100 cm depths). As no snow occurs in the VRB, terrestrial water storage is calculated per grid cell by summing up the surface water storage on impervious areas and all subsurface water storage (i.e.

reservoirs generating soil moisture, baseflow and interflow). The model is run at a daily time step with a spatial discretization of 0.25° (~28 km at the equator).

The modelling experiment covers the period 2000-2012 with 3-year model warm-up period (2000-2002), 6 years for model calibration (2003-2008) and 4 years for model evaluation (2009-2012). The model is calibrated and evaluated with the available daily in-situ streamflow datasets from 11 locations (Figure 2a), while the evaluation with satellite datasets of evaporation, soil

moisture and terrestrial water storage is done at a monthly time step to avoid the impact of mismatches in the daily data retrieval periods among the satellite data sources.

### 2.6 Multisite model calibration on streamflow data

A multisite calibration strategy is adopted by simultaneously constraining the model with the 11 streamflow ($Q$) gauging

stations (Figure 2) to infer a unique parameter set for the whole basin. The multi-objective Kling-Gupta efficiency ($E_{KG}$) (Kling et al., 2012) is used for the formulation of the objective function $\Phi_Q$, which has to be minimized and is formulated as follows:



$$\Phi_Q = 1 - \left[\frac{1}{g}\sum_{i=1}^{g} E_{KG,i}\left(Q_{mod,i}, Q_{obs,i}\right)\right], \text{ with} \tag{3}$$

$$E_{KG} = 1 - \sqrt{(r_{KG} - 1)^2 + (\beta_{KG} - 1)^2 + (\gamma_{KG} - 1)^2} \tag{4}$$

Where $g$ is the number of gauging stations, $r_{KG}$ is the Pearson correlation coefficient, $\beta_{KG}$ is the bias term (i.e. the ratio of the means), and $\gamma_{KG}$ is the variability term (i.e. the ratio of the coefficients of variation) between the observed ($Q_{obs}$) and modelled ($Q_{mod}$) streamflow, with $\mu$ and $\sigma$ representing the mean and the standard deviation. The $E_{KG}$ ranges from negative infinity to

its optimal value that is unity. As a reference, $E_{KG} > -0.41$ indicates that the model is better than the mean observed flow (Knoben et al., 2019). $\Phi_Q$ ranges from its optimal value that is 0 to positive infinity.

The model is calibrated solely with $Q$ data because it is the only available in-situ measurement, and to avoid potential trade-offs of a multivariate calibration that would result in difficulties in identifying the source of variation in the model performance (i.e. input data vs. model parametrization). The parameter estimation is done with the dynamically dimensioned search

algorithm (Tolson and Shoemaker, 2007) using 4,000 iterations for each of the 102 rainfall-temperature dataset combinations.

### 2.7 Multivariable model evaluation with streamflow and satellite data

In addition to $Q$, several non-commensurable and satellite-based variables are used for model evaluation (Table 2). The model performance for $Q$ is evaluated with $E_{KG}$. The bias-insensitive Pearson's correlation coefficient ($r$) is used to assess the

temporal dynamics of $S_t$, $S_u$ and $E_a$ because the model is not calibrated on these variables, and their evaluation datasets are satellite-derived products that encompass uncertainties and can be biased.

The spatial pattern representation of hydrological processes is assessed by using a bias-insensitive and multi-component metric developed by Dembélé et al. (2020). The proposed spatial pattern efficiency ($E_{SP}$) metric is formulated similarly to the $E_{KG}$ (Equation 4) but it focuses only on the spatial pattern of variables rather than on their absolute values (like the SPAEF; Koch

et al., 2018). $E_{SP}$ simultaneously assesses the dynamics, the spatial variability, and the locational matching of grid cells between the observed ($X_{obs}$) and modelled ($X_{mod}$) variables. Considering two variables $X_{obs}$ and $X_{mod}$ composed of $n$ cells, $E_{SP}$ is defined as follows:

$$E_{SP} = 1 - \sqrt{(r_s - 1)^2 + (\gamma - 1)^2 + (\alpha - 1)^2}, \text{ with} \tag{5}$$

$$r_s = 1 - \frac{6\sum_1^n d_i^2}{n(n^2 - 1)}, \tag{6}$$

$$\gamma = \frac{\frac{\sigma_{mod}}{\mu_{mod}}}{\frac{\sigma_{obs}}{\mu_{obs}}} \text{ and} \tag{7}$$

$$\alpha = 1 - E_{RMS}\left(Z_{X_{mod}}, Z_{X_{obs}}\right) \tag{8}$$

where $r_s$ is the Spearman rank-order correlation coefficient with $d_i$ the difference between the ranks of the $i^{th}$ cell of $X_{mod}$ and $X_{obs}$. $\gamma$ is the variability ratio (i.e. the ratio of the coefficients of variation) that assesses the similarity in the dispersion of the

probability distributions of $X_{mod}$ and $X_{obs}$, with $\mu$ and $\sigma$ representing the mean and the standard deviation, and $\alpha$ the spatial



location matching term calculated as the root mean squared error ($E_{RMS}$) of the standardized values (z-scores, $Z_X$) of $X_{mod}$ and $X_{obs}$ (Dembélé et al., 2020). $E_{SP}$ ranges from negative infinity to 1, which is its optimal value. $E_{SP}$ does not have an inherent benchmark, also like $E_{KG}$ (Knoben et al., 2019). For $E_{SP} = 0$, the ranks of the observed and modelled variables are moderately related (i.e. $r_s = 0.55$), while no association among the ranks (i.e. $r_s = 0$) results in $E_{SP} = -0.67$ (cf. Supplementary Material of
Dembélé et al., 2020). However, the main point of using $E_{SP}$ here is not to strictly conclude how well the modelled spatial patterns reproduce the observed patterns, otherwise a benchmark should be used (Schaefli and Gupta, 2007;Seibert et al., 2018), but rather to determine if a modelled spatial pattern is better than another. The spatial pattern evaluation is completed for $S_u$ and $E_a$, while only the temporal dynamics of $S_t$ are assessed due to the coarse spatial resolution of the GRACE data.

The relative variation in model performance is assessed with the second-order coefficient of variation ($V_2$) (Kvålseth, 2017). $V_2$ is an alternative to the classic Pearson's coefficient of variation ($V$), which has significant limitations that are comprehensively discussed by Kvålseth (2017). For all sample data $x = (x_1,…, x_n) \in R^n$, with $R = (-∞, ∞)$, $V_2$ is defined as follows:

$$V_2 = \left( \frac{s^2}{s^2 + \bar{x}^2} \right)^{1/2} \tag{9}$$

where $s$ is the standard deviation and $\bar{x}$ is the mean of $x$. $V_2$ varies from 0 to 1 or 0% to 100%, and represents the distance
between $x$ and $\bar{x}$ relative to the distance between $x$ and the origin zero.

## 3 Results

The results are presented and discussed for the entire simulation period (2003-2012, i.e. combined calibration and evaluation periods) because reliable meteorological datasets are expected to produce a plausible representation of hydrological processes
independently of the modelling period (Bisselink et al., 2016). Separated results are provided for the calibration and evaluation periods in the Supporting Information (SI).

### 3.1 Model performance for streamflow

For daily streamflow ($Q$), all input dataset combinations show a median $E_{KG} > 0.5$, except those having JRA-55 as rainfall
input (Figure 3); this can be justified by the coarse spatial resolution of that product. The ranking of the rainfall and temperature datasets based on the model performance for $Q$ is provided in Appendix A3. The analysis of model performance for $Q$ is done for the entire VRB and not per climatic zone due to the limited number of stations. As expected, the discrepancies in median $E_{KG}$ are more pronounced across rainfall datasets than across temperature datasets, as visible in the color-coded ranking of the products in Figure 3. For a given rainfall product, the ranking among all rainfall products hardly varies with different
temperature products. The ranking of all the datasets for the model performance for $Q$ is also summarized in Appendix A3. The overall stronger impact of the choice of the rainfall dataset on $E_{KG}$ of $Q$ becomes also clear from the second-order





coefficient of variations ($V_2$) of the median $E_{KG}$ (Table S1 in SI). For rainfall datasets, the $V_2$ across temperature datasets varies between 0.5% for GSMaP-std and 4% for JRA-55, with an average $V_2$ of 2%. For temperature datasets, the $V_2$ of median $E_{KG}$ of $Q$ across rainfall datasets varies between 10% for MERRA-2 and 12% for ERA5, with an average $V_2$ of 11%. This result

suggests that the choice of a rainfall dataset has a stronger impact on the $E_{KG}$ of $Q$ than the choice of a temperature dataset.

The analysis of the components of $E_{KG}$ (i.e. the Pearson correlation $r_{KG}$, the bias $\beta_{KG}$ and the variation $\gamma_{KG}$) reveals that, when choosing a rainfall dataset, there is more uncertainty in the bias of $Q$ ($V_2 = 14\%$) than in its variability ($V_2 = 6\%$) and in its dynamics ($V_2 = 3\%$), which is in agreement with the work of Thiemig et al. (2013). Detailed results on the performance for $Q$ (i.e. $E_{KG}$, $r_{KG}$, $\beta_{KG}$ and $\gamma_{KG}$) and the ranking of the datasets with separate results for the calibration and evaluation periods are

provided in the SI (Tables S1-S12, Figures S7-S11).

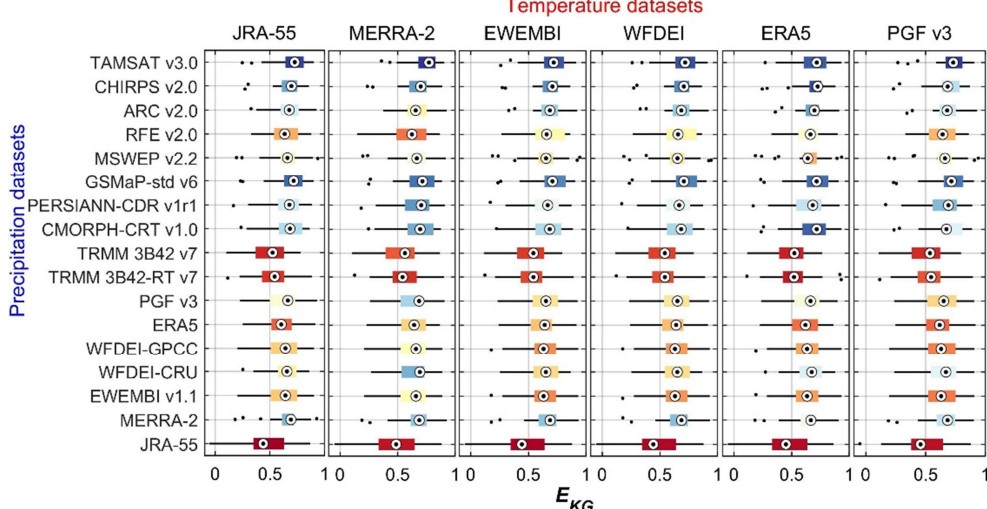

**Figure 3. Kling-Gupta efficiency ($E_{KG}$) of daily streamflow ($Q$) over the simulation period (2003-2012) for 102 combinations of 17 rainfall datasets (y-axis) and 6 temperature datasets (subplots on x-axis) used as inputs in the mHM model. Each boxplot has 22 values representing the combined performance for the calibration and evaluation periods for 11 streamflow gauging stations. The**

**boxplots are colored from the best (blue) to the worst performance (red) based on the median value.**

### 3.2 Model performance for terrestrial water storage

The model performance for the temporal dynamics of monthly terrestrial water storage ($S_t$) compared to the GRACE product is shown in Figure 4 (see the SI for monthly time series, Figures S23-S27). The average Pearson correlation coefficient ($r$) of

$S_t$ for all datasets in the entire VRB is 0.80, with discrepancies across climatic zones. The driest and wettest climatic zones show the lowest performances, i.e. Sahelian ($r = 0.67$) and Guinean ($r = 0.60$) zones, compared to the intermediate climatic





zones, i.e. Sudano-Sahelian ($r = 0.72$) and Sudanian ($r = 0.79$) zones. Appendix A3 provides the ranking of all the meteorological datasets for the model performance for $S_t$.

The rainfall datasets show different performances across climatic zones, with ARC showing the highest score for all the

climatic zones except the Guinean zone, where CMORPH-CRT ranks first. The choice of the rainfall dataset leads to an average $V_2$ of 15% for the $r$ of $S_t$, while the average $V_2$ is 5% for the choice of the temperature dataset. Detailed results are provided in the SI (Tables S13, Figures S12-S22).

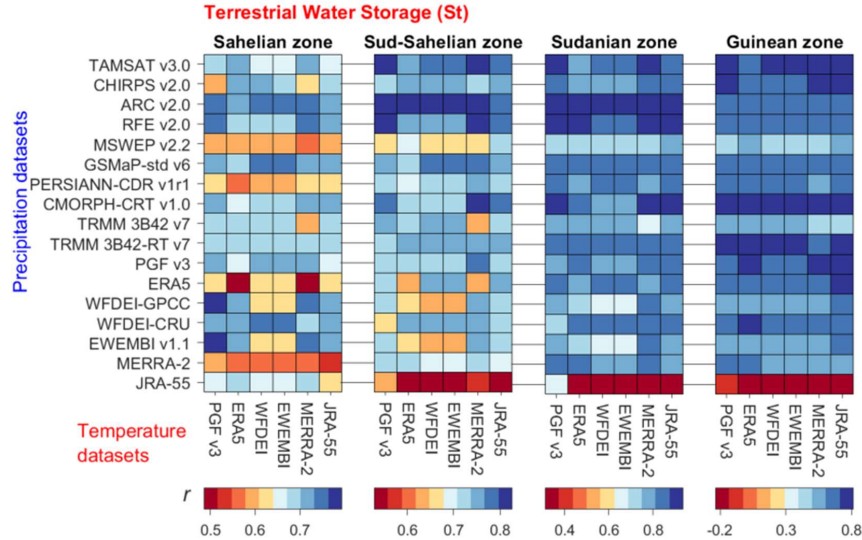

**Figure 4. Pearson correlation coefficient ($r$) of modelled terrestrial water storage compared to GRACE data in four climatic zones**
**in the Volta River basin over the simulation period (2003-2012) considering 102 combinations of rainfall (y-axis) and temperature datasets (subplots on x-axis) used as forcing for the hydrological model.**

### 3.3 Model performance for soil moisture

Figure 5 shows the model performance for the temporal dynamics of monthly soil moisture ($S_u$) compared to the ESA CCI

product (see the SI for monthly time series, Figures S39-S43). The average $r$ of $S_u$ for the entire VRB over all datasets is 0.93. The $r$ of $S_u$ decreases from the drier to the wetter climatic zones: Sahelian ($r = 0.94$), Sudano-Sahelian ($r = 0.94$), Sudanian ($r = 0.92$) and Guinean ($r = 0.86$). The ranking of the meteorological datasets based on the model performance for $S_u$ is provided in Appendix A3. EWEMBI and WFDEI-GPCC show the highest performance in the Sahelian and Suadano-Sahelian zones respectively, while MERRA-2 shows the highest performance in the Sudanian and Guinean zones. The choice of the rainfall

dataset leads to an average $V_2$ of 4% for the temporal dynamics of $S_u$, while the average $V_2$ is 2% for the choice of the temperature dataset.



The spatial patterns of $S_u$ show considerable differences when using different combinations of rainfall and temperature input datasets, as illustrated in Figure 6 (see similar maps for all the meteorological datasets in the SI, Figures S44-S45). The south-north gradient of increasing aridity is not similarly spread among the rainfall-temperature dataset combinations. More

interestingly, west-east differences in the spatial patterns of $S_u$ can be observed. These differences in spatial pattern reproduction can also be seen in the spatial pattern efficiency metric ($E_{SP}$) of $S_u$ for the 102 rainfall-temperature dataset combinations (Figure 7). The average $E_{SP}$ of $S_u$ in the VRB over all datasets is -0.11.

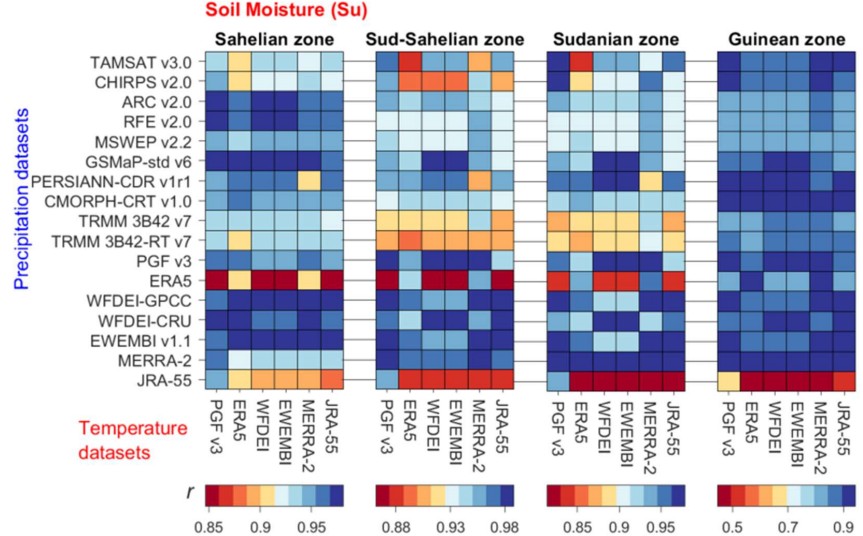

**Figure 5. Pearson correlation coefficient ($r$) of modelled soil moisture ($S_u$) compared to ESA CCI data over the simulation period (2003-2012) considering 102 combinations of rainfall (y-axis) and temperature datasets (subplots on x-axis) used as forcing for the hydrological model.**

For the entire VRB, the choice of the rainfall dataset leads to an average variation of 61% for the $E_{SP}$ of $S_u$, while the choice

of the temperature dataset involves a variation of 45%. Lower impacts of data choices are observed in the climatic zones where the climate is homogeneous as compared to the entire VRB. The choice of a rainfall dataset is more critical for the $E_{SP}$ of $S_u$ in the driest and wettest climatic zones, i.e. Sahelian ($E_{SP}$ = -0.47, $V_2$ = 25%) and Guinean ($E_{SP}$ = -0.40, $V_2$ = 26%) zones, than the intermediate zones, i.e. Sudano-Sahelian ($E_{SP}$ = -0.37, $V_2$ = 11%) and Sudanian ($E_{SP}$ = -0.39, $V_2$ = 17%) zones. A smaller impact on the $E_{SP}$ of $S_u$ is observed for the choice of the temperature dataset: Sahelian ($V_2$ = 8%), Guinean ($V_2$ = 19%), Sudano-

Sahelian ($V_2$ = 5%) and Sudanian ($V_2$ = 9%) zones. Detailed results on the model performance for $S_u$ and the ranking of the datasets for the calibration and evaluation periods are provided in the SI (Tables S14-S15, Figures S28-S38).

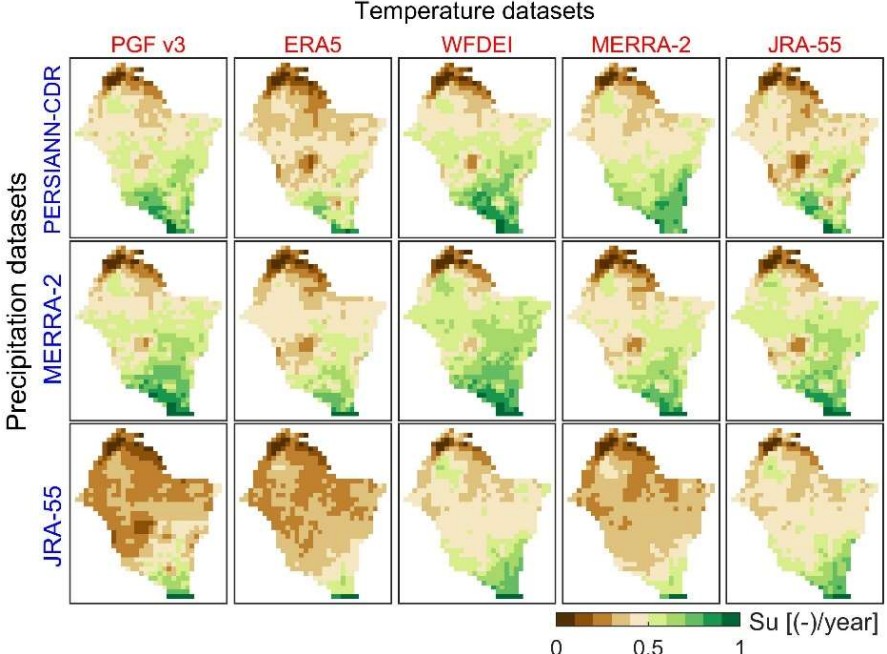

**Figure 6. Maps of long-term (2003-2012) average of annual soil moisture ($S_u$) obtained with different forcing of rainfall (y-axis, blue font) and temperature (x-axis, red font) datasets. The values are normalized between 0 and 1 to emphasize spatial patterns and to use a unique color scale.**




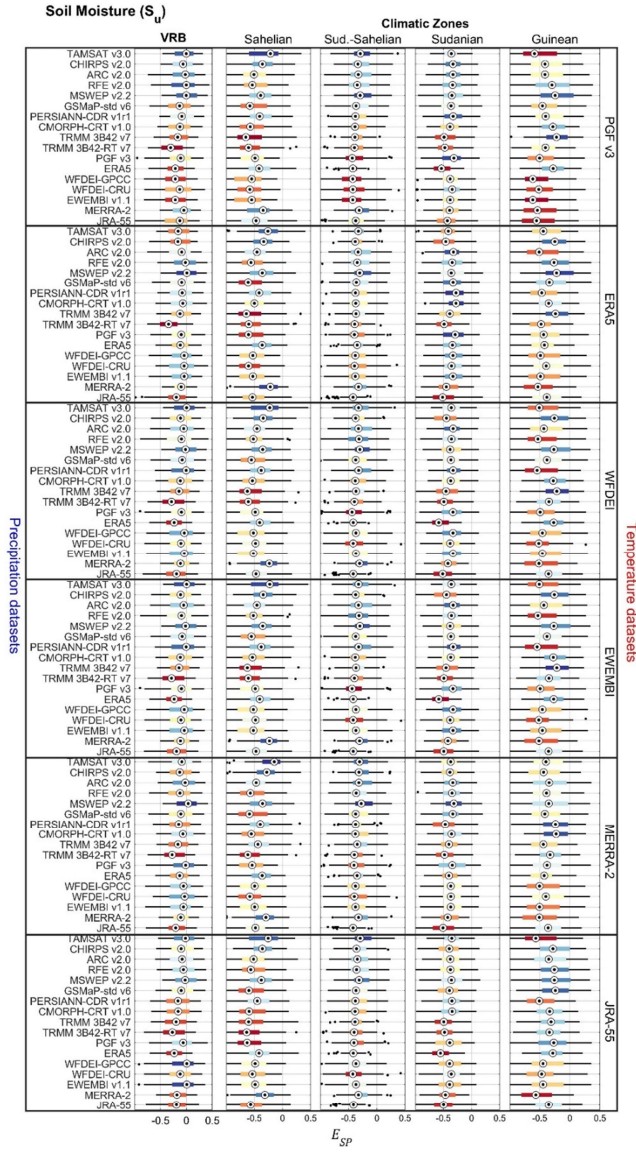

**Figure 7.** Spatial pattern efficiency ($E_{SP}$) of soil moisture ($S_u$) over the entire simulation period (2003-2012) for the Volta River basin
(VRB) and its climatic zones, using different combinations of precipitation and temperature datasets used as input for hydrological
modelling. Each boxplot has 120 values corresponding to the number of months. The boxplots are colored from the best (blue) to
the worst performance (red) based on the median value.





### 3.4 Model performance for actual evaporation

The model performance for the temporal dynamics of monthly actual evaporation ($E_a$) compared to the GLEAM product is shown in Figure 8 (see the SI for monthly time series, Figures S57-S61). The average $r$ of $E_a$ for the entire VRB over all

datasets is 0.93. Similarly to $S_u$, the $r$ of $E_a$ is higher is the driest climatic zones: Sahelian ($r = 0.94$), Sudano-Sahelian ($r = 0.94$), Sudanian ($r = 0.89$) and Guinean ($r = 0.81$). However, the predictive skill of the model for the temporal dynamics of $E_a$ is higher than its predictive skill for $E_a$ in the wetter climatic zones. Appendix A3 shows the ranking of all the meteorological datasets for the model performance for $E_a$. The rainfall datasets show different performances across climatic zones, with the following best datasets: PERSIANN-CDR in the Sahelian zone, EWEMBI and WFDEI-GPCC in the Soudano-Sahelian zone,

ARC in the Sudanian and Guinean zones. The choice of the rainfall dataset leads to an average $V_2$ of 4% for the temporal dynamics of $E_a$, while the average $V_2$ is 1.5% for the choice of the temperature dataset, which aligns with the findings of Jung et al. (2019).

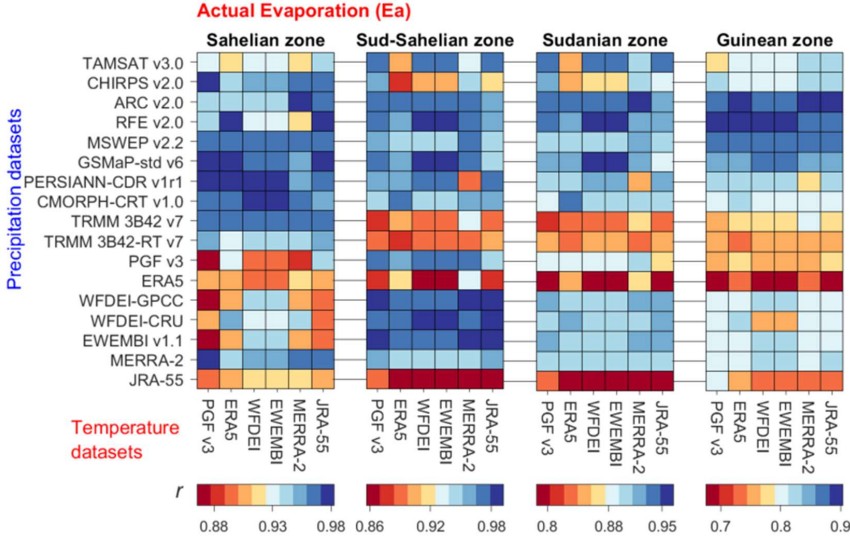

**Figure 8. Pearson correlation coefficient ($r$) of modelled actual evaporation ($E_a$) compared to GLEAM data over the simulation**
**period (2003-2012) considering 102 combinations of rainfall (y-axis) and temperature datasets (subplots on x-axis) used as forcing for the hydrological model.**

As for $S_u$, the choice of input datasets has a considerable impact on the reproduction of the spatial patterns of $E_a$ (Figure 9). Similar maps for all the meteorological datasets are provided in the SI (Figures S62-S63). It can be observed that different

rainfall-temperature combinations used to force the model result in large discrepancies in the spatial pattern of $E_a$, especially in the southern region. The south-north gradient of increasing aridity with west-east differences is represented differently



among the rainfall-temperature dataset combinations (see e.g., the difference between the first two columns of the first row in Figure 9)

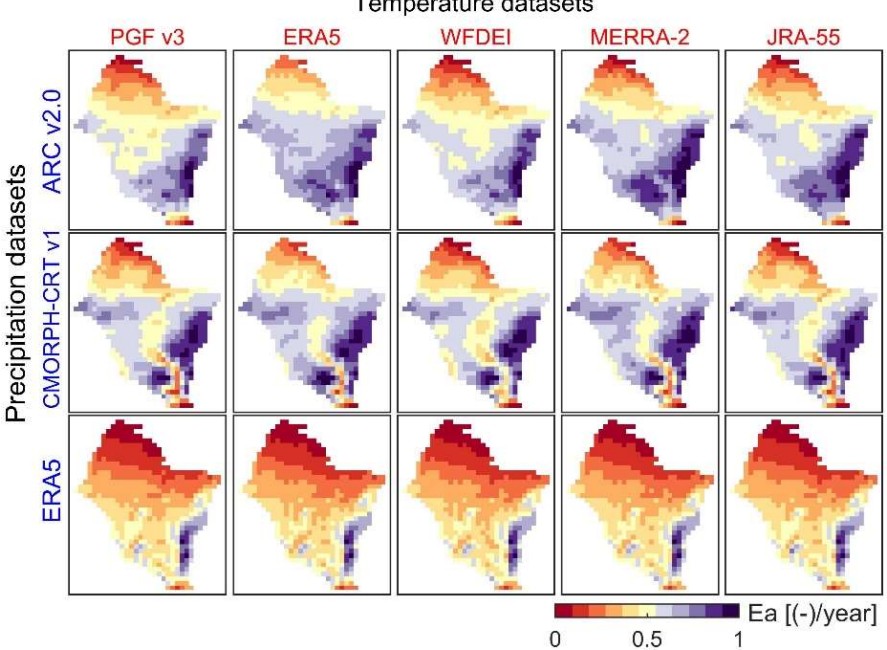

**Figure 9. Maps of long-term (2003-2012) average of annual actual evaporation ($E_a$) obtained with different forcing of rainfall (y-axis, blue font) and temperature (x-axis, red font) datasets. The values are normalized between 0 and 1 to emphasize spatial patterns and to use a unique color scale.**

The $E_{SP}$ of $E_a$ for the 102 rainfall-temperature dataset combinations in the VRB and the climatic zones is given in Figure 10. The average $E_{SP}$ of $E_a$ in the VRB over all datasets is 0.07, which is higher than for $S_u$ ($E_{SP}$ = -0.11). The choice of the rainfall dataset for the VRB affects the $E_{SP}$ of $E_a$ on average by 93%, while the choice of the temperature dataset involves a variation 33%. However, lower impacts of data choices are observed in the climatic zones. The choice of rainfall dataset is more critical for the $E_{SP}$ of $E_a$ in the driest and wettest climatic zones, i.e. Sahelian ($E_{SP}$ = -0.99, $V_2$ = 49%) and Guinean ($E_{SP}$ = -0.79, $V_2$ = 37%) zones, than the intermediate zones, i.e. Sudano-Sahelian ($E_{SP}$ = -0.35, $V_2$ = 36%) and Sudanian ($E_{SP}$ = -0.42, $V_2$ = 49%) zones. A smaller impact on the $E_{SP}$ of $E_a$ is observed for the choice of the temperature dataset: Sahelian ($V_2$ = 21%), Guinean ($V_2$ = 10%), Sudano-Sahelian ($V_2$ = 17%) and Sudanian ($V_2$ = 21%) zones. Detailed results on the model performance for $E_a$ and the ranking of the datasets for the calibration and evaluation periods are provided in the SI (Tables S16-S17, Figures S46-S56).



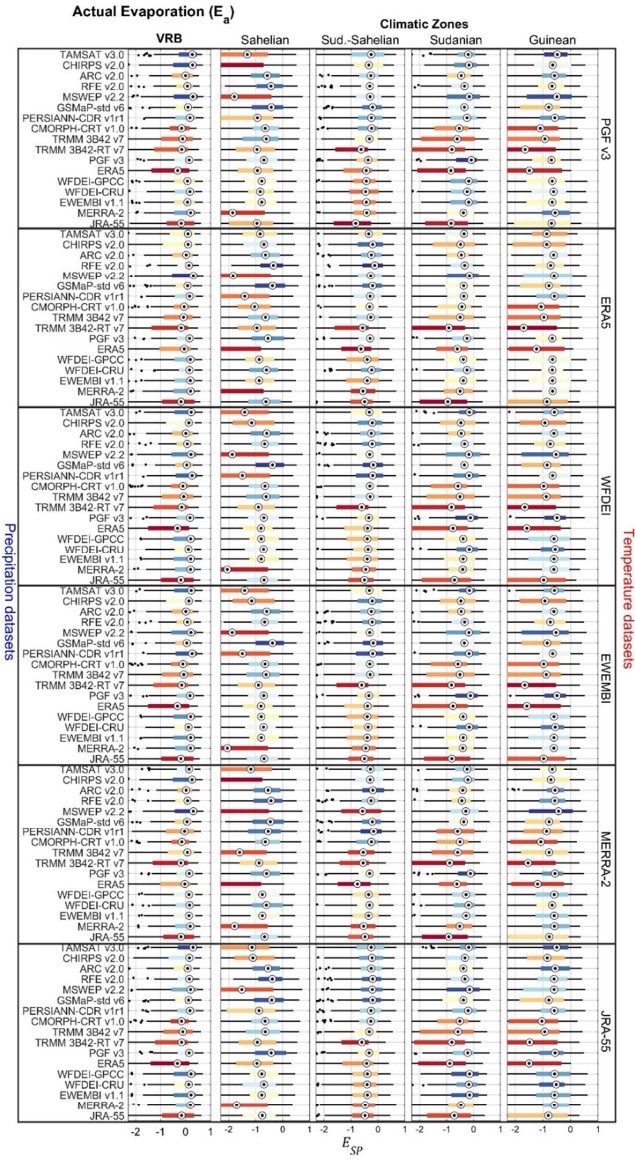

**Figure 10. Spatial pattern efficiency ($E_{SP}$) of actual evaporation ($E_a$) over the entire simulation period (2003-2012) for the Volta River basin (VRB) and its climatic zones, using different combinations of precipitation and temperature datasets used as input for hydrological modelling. Each boxplot has 120 values corresponding to the number of months. The boxplots are colored from the best (blue) to the worst performance (red) based on the median value.**





## 4 Discussions

This study builds upon and expands existing research studies on the evaluation of meteorological datasets in several ways:

420        (i) The evaluation of the spatial patterns of multiple hydrological processes (i.e. streamflow, actual evaporation, soil moisture, and terrestrial water storage) in addition to the more classically evaluated temporal dynamic.

       (ii) The evaluation of a high number of both satellite-based and reanalysis rainfall datasets considered in combination with different temperature datasets.

       (iii) The assessment of the model performance across four considerably different climatic zones from semi-arid to sub-

425        humid.

The overall outcome of this analysis is the ranking of all the meteorological datasets based on their ability to simulate various hydrological processes across different climatic zones in the VRB (Appendix A3). It is worth noting that the overall ranking shows which product is best or worst at simulating a given hydrological flux or state variable. However, the ranking does not systematically tell whether a dataset is good or bad. Only the skill scores can be used to draw a judgement on the adequacy of

a given dataset to produce plausible model outputs.

The results show that there is no single rainfall dataset outperforming the others in reproducing all hydrological processes across different climatic zones. These findings align with previous studies in the sense that there is no rainfall dataset that is the best everywhere (Beck et al., 2017b;Sylla et al., 2013). For datasets providing both rainfall and temperature data, the combination of the two variables as model input is not necessarily the best option for obtaining the highest performance in

modelling a given hydrological state or flux variable. The best rainfall-temperature combinations for the spatiotemporal representation of each hydrological flux and state variable are provided in the SI (Figure S7).

The results can be considered valid for West Africa and regions with similar hydroclimatic and physical features. A wider generalization of the findings should be done with caution and after repeating similar evaluation studies in other places. Nevertheless, the key message is that: *"there is no rainfall dataset of all hydrological processes"* and *"the best rainfall dataset*

*for temporal dynamics might not be the best for spatial patterns"*.

Despite the efforts to produce a comprehensive evaluation of the meteorological datasets, the results obtained might be subject to uncertainties related to the potential model structural deficiencies as well as errors in the observational datasets used for the model evaluation (McMillan et al., 2010;Renard et al., 2010;Gupta and Govindaraju, 2019). The distribution of the final model parameters (Figures S65-S66) highlights the possibility of obtaining equally good model performances for different parameter

sets (i.e. equifinality), which can be a justification for model recalibration. A detailed analysis of parameter variability as a function of input data is beyond the scope of the current study, but could build the basis of future research, namely to identify data errors by analyzing parameter patterns (e.g. rooting depth), and resolve potential structural deficiencies of the mHM model. However, the mHM is chosen because of its adequacy for the experiment of this study (for model selection, see Addor and Melsen, 2019). The structure of mHM allows the representation of seamless spatial patterns of hydrological processes

through the MPR scheme (Samaniego et al., 2017). In addition, mHM facilitates parameter regionalization and is therefore


convenient for large-scale modelling, and it harnesses the full potential of the forcing datasets as it is a fully distributed model that has performed well in previous studies including those in the VRB (e.g. Poméon et al., 2018;Dembélé et al., 2020). Regarding the model evaluation, the comparison between the observed and modelled hydrological processes is done only on their temporal dynamics and spatial patterns using bias-insensitive metrics, except for streamflow, which limits the potential
impact of satellite data uncertainty.

The model is calibrated only on $Q$ data despite the known limitations of the $Q$-only calibration (Demirel et al., 2018). However, regarding the goal of this study, that was the best option to obtain the impact of various meteorological forcing datasets on the plausibility of hydrological processes. As no rainfall dataset ranks first in simulating all the hydrological processes, this study confirms that model calibration on multiple variables is a way forward in improving the overall representation of the
hydrological system and increasing the predictive skill of hydrological models (Dembélé et al., 2020;Dembélé et al., in review). The domain-wide calibration strategy adopted in this study generates a unique parameter set for the simulation of multiple hydrological processes across several catchments with different hydroclimatic features, which has the consequence of having local differences in model performance. However, domain-wide calibration has proved to perform similarly to domain-split calibration in previous studies (Mizukami et al., 2017), and it was ideal for this study because of the interest in simulating
seamless spatial patterns, which might have not been possible with separately simulated portions of the basin. Moreover, the main goal of this study is to assess the adequacy of the meteorological datasets for large-scale hydrological modelling, knowing that these datasets usually have a coarse spatial resolution with pixels often averaged over regions with strong sub-grid variability.

Finally, the importance of regional evaluation is emphasized by this study because some region-tailored datasets (e.g.
TAMSAT and ARC) which are not included in global scale studies (e.g. Beck et al., 2017b;Mazzoleni et al., 2019;Essou et al., 2016) outperform global datasets. The decision to use a given dataset is not only motivated by the availability or the accuracy of the data, but also by data accessibility (i.e. storage platforms, openness, format, pre-processing requirement, etc). The findings of this study provide further awareness for the data users and improvement avenues for data producers in their quest of the most accurate products.

**5 Conclusion**

This modelling study evaluates the ability of multiple combinations of rainfall-temperature datasets to reproduce plausible hydrological processes and patterns. The experiment is done in the Volta River basin with the fully distributed mesoscale Hydrologic Model (mHM) over a 10-year period (2003-2012), using 17 rainfall and 6 temperature datasets from satellite and reanalysis sources. The spatial and temporal representation of streamflow, terrestrial water storage, soil moisture and actual
evaporation are evaluated using in-situ and satellite remote sensing observational datasets. The key findings are:



- No rainfall dataset consistently outperforms all the others in reproducing the highest model performance for all hydrological processes, and the best dataset for the temporal dynamics is not necessarily the best for the spatial patterns.

- Rainfall datasets have a higher impact on the spatiotemporal representation of hydrological processes than temperature datasets, but the later have a higher influence on the spatial patterns of soil moisture.

- The large-scale performance for the meteorological datasets is not always valid for sub-regions in the same basin.

The findings of this study give a critical insight of the performance for several meteorological datasets in the challenging hydroclimatic environment of West Africa. They are expected to foster further research initiatives in improving the gridded meteorological datasets and further draw users' attention on the contrasting performances of these datasets in modelling

hydrological fluxes and state variables. Efforts should be devoted in reporting on the impact of data uncertainties on process representation in hydrological modelling, especially when model outputs are used for decision-making.

Future studies can test the transferability of the model's global parameters across different input datasets, i.e. how reliable a parameter set obtained with a given input dataset is for running the same model with a different input dataset. The answer to this research question will shed light on the necessity of model recalibration when using different meteorological forcing.

Furthermore, the predictive skill of the model can be improved with a parameter sensitivity analysis to determine parameters that affect the spatiotemporal representation of each hydrological flux and state variable.





## 6 Appendix A: Figures

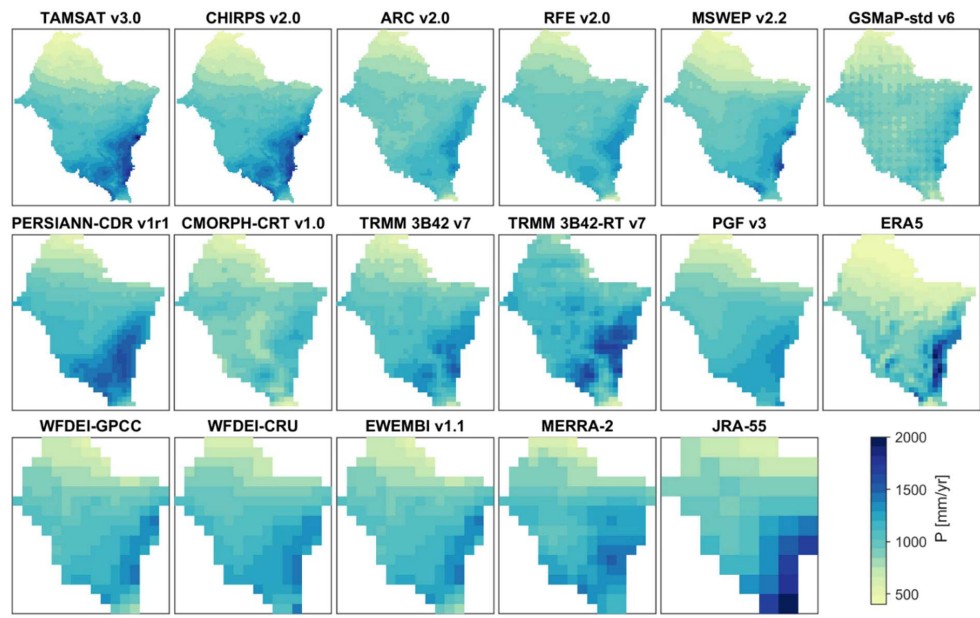

**A1. Mean annual rainfall totals over the period 2003-2012 for 17 rainfall datasets the Volta River basin**



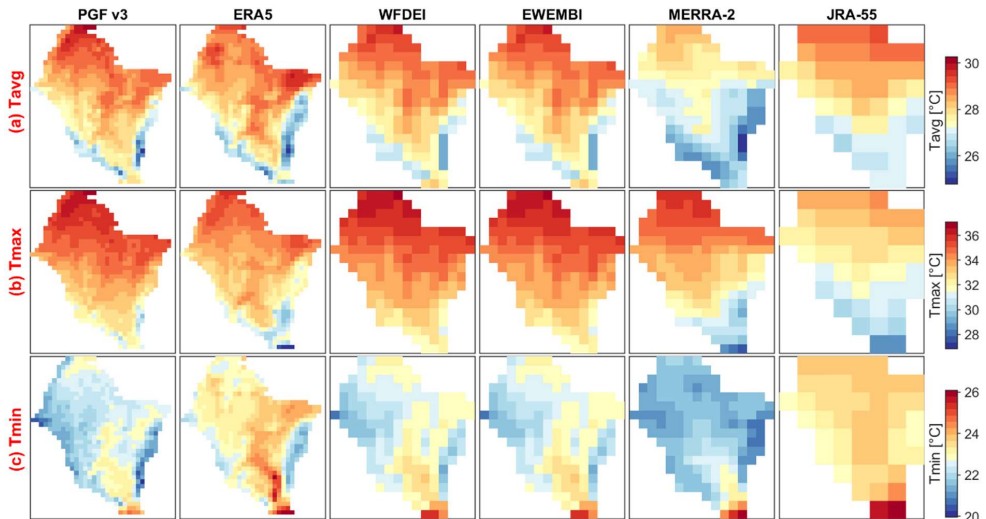

**A2. Mean annual air temperature (average (a), maximum (b) and minimum (c)) over the period 2003-2012 for 6 temperature datasets in the Volta River basin**


| | | **Skill scores** For temporal dynamics: $E_{KG}$ of $Q$, $r$ of $St$, $Su$ and $Ea$ \| For spatial patterns: $E_{SP}$ of $Su$ and $Ea$ | | | | | | | | | | | | | | | | | | | | | | | | |
|---|---|---|---|---|---|---|---|---|---|---|---|---|---|---|---|---|---|---|---|---|---|---|---|---|---|---|---|
| | | **VRB** | | | | | | **Sahelian zone** | | | | | **Sudano-Sahelian zone** | | | | | **Sudanian zone** | | | | | **Guinean zone** | | | | |
| | | Temporal dynamics | | | | Spatial patterns | | Temporal dynamics | | | Spatial patterns | | Temporal dynamics | | | Spatial patterns | | Temporal dynamics | | | Spatial patterns | | Temporal dynamics | | | Spatial patterns | |
| | | Q | St | Su | Ea | Su | Ea | St | Su | Ea | Su | Ea | St | Su | Ea | Su | Ea | St | Su | Ea | Su | Ea | St | Su | Ea | Su | Ea |
| Rainfall datasets | TAMSAT v3.0 | 0.73 | 0.86 | 0.94 | 0.94 | -0.04 | 0.21 | 0.69 | 0.93 | 0.93 | -0.22 | -1.21 | 0.78 | 0.94 | 0.95 | -0.31 | -0.29 | 0.86 | 0.93 | 0.92 | -0.37 | -0.22 | 0.74 | 0.90 | 0.81 | -0.50 | -0.61 |
| | CHIRPS v2.0 | 0.70 | 0.86 | 0.93 | 0.92 | -0.11 | 0.18 | 0.67 | 0.92 | 0.96 | -0.34 | -1.60 | 0.74 | 0.92 | 0.92 | -0.36 | -0.24 | 0.82 | 0.92 | 0.88 | -0.41 | -0.36 | 0.71 | 0.89 | 0.82 | -0.32 | -0.82 |
| | ARC v2.0 | 0.68 | 0.91 | 0.93 | 0.97 | -0.05 | 0.03 | 0.73 | 0.97 | 0.95 | -0.48 | -0.58 | 0.80 | 0.95 | 0.97 | -0.33 | -0.44 | 0.92 | 0.93 | 0.94 | -0.34 | -0.44 | 0.67 | 0.84 | 0.89 | -0.42 | -0.59 |
| | RFE v2.0 | 0.65 | 0.88 | 0.92 | 0.96 | -0.07 | 0.10 | 0.71 | 0.97 | 0.95 | -0.55 | -0.49 | 0.78 | 0.94 | 0.98 | -0.34 | -0.22 | 0.89 | 0.91 | 0.94 | -0.35 | -0.35 | 0.65 | 0.83 | 0.89 | -0.38 | -0.67 |
| | MSWEP v2.2 | 0.66 | 0.72 | 0.92 | 0.94 | 0.00 | 0.26 | 0.58 | 0.94 | 0.96 | -0.36 | -1.90 | 0.68 | 0.94 | 0.95 | -0.30 | -0.32 | 0.73 | 0.91 | 0.90 | -0.36 | -0.19 | 0.49 | 0.82 | 0.86 | -0.27 | -0.53 |
| | GSMaP-std v6 | 0.71 | 0.84 | 0.94 | 0.95 | -0.10 | 0.08 | 0.72 | 0.97 | 0.97 | -0.58 | -0.40 | 0.76 | 0.96 | 0.97 | -0.37 | -0.19 | 0.84 | 0.94 | 0.92 | -0.36 | -0.36 | 0.66 | 0.88 | 0.86 | -0.37 | -0.81 |
| | PERSIANN-CDR v1r1 | 0.68 | 0.80 | 0.96 | 0.94 | -0.08 | 0.17 | 0.59 | 0.95 | 0.97 | -0.41 | -0.22 | 0.76 | 0.96 | 0.95 | -0.36 | -0.22 | 0.81 | 0.95 | 0.90 | -0.35 | -0.29 | 0.64 | 0.92 | 0.82 | -0.45 | -0.65 |
| | CMORPH-CRT v1.0 | 0.69 | 0.87 | 0.94 | 0.94 | -0.11 | -0.12 | 0.69 | 0.95 | 0.97 | -0.55 | -0.72 | 0.76 | 0.94 | 0.96 | -0.38 | -0.28 | 0.86 | 0.93 | 0.90 | -0.35 | -0.53 | 0.78 | 0.92 | 0.81 | -0.29 | -1.04 |
| | TRMM 3B42 v7 | 0.54 | 0.77 | 0.91 | 0.88 | -0.16 | -0.06 | 0.67 | 0.93 | 0.97 | -0.60 | -0.79 | 0.72 | 0.92 | 0.90 | -0.37 | -0.34 | 0.78 | 0.89 | 0.83 | -0.45 | -0.55 | 0.52 | 0.84 | 0.78 | -0.28 | -0.89 |
| | TRMM 3B42-RT v7 | 0.54 | 0.86 | 0.91 | 0.89 | -0.30 | -0.16 | 0.68 | 0.95 | 0.94 | -0.61 | -0.92 | 0.74 | 0.90 | 0.89 | -0.40 | -0.59 | 0.84 | 0.89 | 0.83 | -0.49 | -0.85 | 0.72 | 0.85 | 0.75 | -0.38 | -1.61 |
| | PGF v3 | 0.66 | 0.82 | 0.96 | 0.93 | -0.08 | 0.16 | 0.69 | 0.95 | 0.90 | -0.54 | -0.61 | 0.73 | 0.97 | 0.97 | -0.42 | -0.33 | 0.82 | 0.95 | 0.89 | -0.33 | -0.15 | 0.72 | 0.90 | 0.76 | -0.43 | -0.58 |
| | ERA5 | 0.63 | 0.82 | 0.91 | 0.87 | -0.20 | -0.23 | 0.57 | 0.87 | 0.90 | -0.40 | -1.42 | 0.70 | 0.89 | 0.89 | -0.40 | -0.50 | 0.81 | 0.87 | 0.81 | -0.50 | -0.75 | 0.69 | 0.87 | 0.70 | -0.33 | -1.43 |
| | WFDEI-GPCC | 0.64 | 0.75 | 0.96 | 0.95 | -0.07 | 0.17 | 0.71 | 0.97 | 0.91 | -0.52 | -0.79 | 0.68 | 0.98 | 0.98 | -0.38 | -0.39 | 0.76 | 0.96 | 0.91 | -0.36 | -0.30 | 0.55 | 0.88 | 0.81 | -0.50 | -0.62 |
| | WFDEI-CRU | 0.67 | 0.83 | 0.96 | 0.94 | -0.09 | 0.14 | 0.72 | 0.97 | 0.93 | -0.54 | -0.69 | 0.73 | 0.97 | 0.98 | -0.41 | -0.37 | 0.83 | 0.97 | 0.91 | -0.37 | -0.20 | 0.70 | 0.90 | 0.79 | -0.47 | -0.59 |
| | EWEMBI v1.1 | 0.64 | 0.75 | 0.96 | 0.95 | -0.07 | 0.17 | 0.71 | 0.97 | 0.91 | -0.52 | -0.79 | 0.68 | 0.98 | 0.98 | -0.38 | -0.39 | 0.76 | 0.96 | 0.91 | -0.36 | -0.30 | 0.55 | 0.88 | 0.81 | -0.50 | -0.62 |
| | MERRA-2 | 0.68 | 0.80 | 0.97 | 0.93 | -0.11 | 0.20 | 0.56 | 0.93 | 0.96 | -0.27 | -2.00 | 0.71 | 0.97 | 0.95 | -0.32 | -0.48 | 0.81 | 0.97 | 0.90 | -0.43 | -0.45 | 0.61 | 0.93 | 0.81 | -0.53 | -0.60 |
| | JRA-55 | 0.45 | 0.38 | 0.83 | 0.84 | -0.18 | -0.18 | 0.66 | 0.90 | 0.91 | -0.50 | -0.73 | 0.56 | 0.89 | 0.87 | -0.41 | -0.55 | 0.38 | 0.84 | 0.80 | -0.49 | -0.83 | -0.19 | 0.50 | 0.75 | -0.40 | -0.85 |
| Temperature datasets | JRA-55 | 0.64 | 0.81 | 0.94 | 0.93 | -0.12 | 0.07 | 0.68 | 0.95 | 0.93 | -0.49 | -0.33 | 0.73 | 0.95 | 0.95 | -0.37 | -0.33 | 0.82 | 0.93 | 0.89 | -0.41 | -0.39 | 0.61 | 0.87 | 0.80 | -0.37 | -0.77 |
| | MERRA-2 | 0.66 | 0.79 | 0.92 | 0.92 | -0.10 | 0.07 | 0.66 | 0.94 | 0.94 | -0.45 | -1.14 | 0.71 | 0.93 | 0.94 | -0.36 | -0.37 | 0.78 | 0.91 | 0.89 | -0.39 | -0.44 | 0.59 | 0.85 | 0.81 | -0.39 | -0.76 |
| | EWEMBI | 0.64 | 0.78 | 0.93 | 0.93 | -0.10 | 0.06 | 0.66 | 0.94 | 0.94 | -0.45 | -0.96 | 0.71 | 0.94 | 0.94 | -0.37 | -0.33 | 0.78 | 0.92 | 0.89 | -0.40 | -0.42 | 0.59 | 0.86 | 0.80 | -0.40 | -0.81 |
| | WFDEI | 0.64 | 0.78 | 0.93 | 0.93 | -0.10 | 0.06 | 0.66 | 0.94 | 0.94 | -0.45 | -0.96 | 0.71 | 0.94 | 0.94 | -0.37 | -0.33 | 0.78 | 0.92 | 0.89 | -0.40 | -0.41 | 0.59 | 0.86 | 0.80 | -0.40 | -0.81 |
| | ERA5 | 0.64 | 0.81 | 0.94 | 0.93 | -0.10 | 0.08 | 0.66 | 0.94 | 0.94 | -0.48 | -1.01 | 0.74 | 0.95 | 0.95 | -0.37 | -0.34 | 0.81 | 0.93 | 0.89 | -0.37 | -0.45 | 0.59 | 0.86 | 0.81 | -0.40 | -0.83 |
| | PGF v3 | 0.64 | 0.81 | 0.93 | 0.92 | -0.12 | 0.06 | 0.67 | 0.94 | 0.94 | -0.48 | -1.01 | 0.73 | 0.94 | 0.94 | -0.37 | -0.38 | 0.81 | 0.92 | 0.89 | -0.38 | -0.39 | 0.62 | 0.85 | 0.81 | -0.43 | -0.78 |





**A3. Model performance for streamflow ($Q$), terrestrial water storage ($S_t$), soil moisture ($S_u$) and actual evaporation ($E_a$) using various rainfall-temperature dataset combinations as model inputs. Each score for a given rainfall product represents the average over individual combinations with 6 temperature datasets, while the score is the average over combinations with 17 rainfall datasets for each temperature dataset. The skill scores of the temporal dynamics are obtained with the Kling-Gupta efficiency ($E_{KG}$) for $Q$ and the Pearson's correlation coefficient ($r$) for $S_t$, $S_u$ and $E_a$. The spatial pattern efficiency ($E_{SP}$) is used to assess the spatial representation of $S_u$ and $E_a$. The skill scores are ranked from the best (blue) to the worst (red). The results are shown for the four climatic zones in the Volta River basin (VRB) over the simulation period (2003-2012).**


*Supplement.* The supplement related to this article is available online at: *to be provided by the journal*

*Data availability.* The meteorological and modelling datasets used in this study are freely available via the web links provided in Table 1 and Table 2. More information on satellite-based precipitation datasets can be found at http://ipwg.isac.cnr.it/. The modelling database is available at *repository to be provided upon acceptance*.


*Author contributions.* MD performed the analyses and drafted the manuscript. All authors contributed to the writing, review and editing process that lead to the final manuscript.

*Competing interests.* The authors declare that they have no conflict of interest.


*Acknowledgements.* We thank the providers of the datasets used in this study (see Table 1 and Table 2). We are grateful to the developers of mHM at CHS/UFZ (Germany) for their open-source model. We thank the providers of the streamflow data obtained from the Volta Basin Authority (VBA), the Direction Générale des Ressources en Eau (DGRE) of Burkina Faso, the Hydrological Services Department (HSD) of Ghana and the Direction Générale de l'Eau et de l'Assainissement (DGEA) of Togo.


*Financial support.* Moctar Dembélé was supported by the Swiss Government Excellence Scholarship (2016.0533 / Burkina Faso / OP), and the Doc.Mobility fellowship (SNF, P1LAP2_178071) of the Swiss National Science Foundation. Bettina Schaefli was supported by a research grant of the Swiss National Science Foundation (SNF, PP00P2_157611).


*Review statement.* This paper was edited by *Editor's Name* and reviewed by *reviewers' name/anonymous*.

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
