# Peer review of "Suitability of 17 rainfall and temperature gridded datasets for largescale hydrological modelling in West Africa"

_Hydrology and Earth System Sciences, 2020_

## Referee Comment (RC1) · Anonymous Referee #1 · 16 May 2020

This is an interesting paper that is reasonably well written. Although the assessment includes a large number of datasets, the study area is relatively small, and the model is not recalibrated for each variable, which has led to some questionable conclusions.

You state that "rainfall datasets have contrasting performances across the four climatic zones present in the VRB, suggesting that, in general, basin-wide hydrological model performance might be misleading and invalid for a smaller spatial domain." What makes you think that your results, which also represent a relatively small spatial domain, are not "misleading and invalid" as well? It is stated that "the results can be considered valid for West Africa and regions with similar hydroclimatic and physical features" which

is highly speculative and likely not true given the variation in precipitation dataset performance and gauge network density. To improve the generalizability of the results, the assessment should be expanded to other regions across Africa or the globe. Alternatively, the abstract and discussion should clearly state that the conclusions and the performance ranking of the datasets are not representative of other regions.

The soil moisture, terrestrial water storage, and actual evaporation assessments were carried out without recalibrating the model and therefore the results for these variables are subject to substantial uncertainty. This is supported by the fact that MSWEP, which was used to force GLEAM, does not exhibit good actual evaporation scores. The model should be recalibrated for each variable.

The word "gauge" is not used in the abstract and the datasets are only classified as either satellite or reanalysis. However, the amount of gauge data incorporated in the datasets may well be the overriding factor in determining the performance, given the good performance of TAMSAT and CHIRPS in terms of streamflow.

Figures 7 and 10 are impossible to interpret, way too much information. Should be condensed.

---

## Author Comment (AC1) · 26 May 2020

**Authors' Reply to Anonymous Referee #1**

**"Suitability of 17 rainfall and temperature gridded datasets for largescale hydrological modelling in West Africa" by Dembélé et al.**

*Preliminary remark: the comment numbering has been introduced by the authors for cross-referencing.*

**Anonymous Referee #1**

1) This is an interesting paper that is reasonably well written. Although the assessment includes a large number of datasets, the study area is relatively small, and the model is not recalibrated for each variable, which has led to some questionable conclusions.

*Response: We thank the referee#1 for the positive overall appreciation of our work. As it can be read from the title, our study focuses on the poorly studied region of West Africa, when it comes to hydrological modelling in general and hydrological evaluation of meteorological datasets in particular. Accordingly, our contribution clearly represents added-value, in terms of regional hydrology as well as in terms of hydrological modelling of semi-arid areas. In fact, independent regional evaluation of globally and regionally available datasets are of key importance for hydrology as they can provide new insights that might not be fully highlighted in global studies.*

*Regarding the general critics on the model calibration, we would like to refer the reader to our detailed answer to comment 4. Below we provide more details and answers to the detailed comments of this reviewer.*

2) You state that "rainfall datasets have contrasting performances across the four climatic zones present in the VRB, suggesting that, in general, basin-wide hydrological model performance might be misleading and invalid for a smaller spatial domain." What makes you think that your results, which also represent a relatively small spatial domain, are not "misleading and invalid" as well?

*Response: We thank the referee#1 for pointing out this potentially misleading statement. We agree with this referee that our statement has some strong wordings, which have led to a different interpretation by the referee#1 than what we intended to say. Our original idea is that the overall model performance in the entire, relatively large modelling domain (from a catchment hydrology perspective, the Volta River Basin, VRB, is indeed a large domain) might not be representative for all subdomains. This is especially the case if the modelling domain extends over multiple climatic zones as in the VRB case. For instance, in global studies, the overall global performance of a rainfall dataset is likely different from its performance in sub-regions such as West Africa (e.g. see Figure 3 in Beck et al. 2017b). Therefore, by "smaller spatial domain" we meant a portion of a large domain under evaluation. We will correct this statement to avoid any misunderstanding in the revised manuscript.*

3) It is stated that "the results can be considered valid for West Africa and regions with similar hydroclimatic and physical features" which is highly speculative and likely not true given the variation in precipitation dataset performance and gauge network density. To improve the generalizability of the results, the assessment should be expanded to other regions across Africa or the globe.

Alternatively, the abstract and discussion should clearly state that the conclusions and the performance ranking of the datasets are not representative of other regions.

*Response: We agree with the referee#1 that this isolated statement can be interpreted as speculative. However, we did not want to imply any certainty but the possibility that the results might be transferable to other places. This is expressed in the sentence following the aforementioned sentence: "A wider generalization of the findings should be done with caution and after repeating similar evaluation studies in other places". Also, we did not intend to generalize our findings to other regions, which is very clear from the mentions "West Africa" in the title, and "Volta River Basin (VRB) in West Africa" in the very first sentence of our manuscript.*

*We agree that we should not have mentioned transferability to other similar climates outside Africa since the performance of any remote sensing-based meteorological data set for hydrological modelling varies across the globe due to many other factors not only related to regional aridity.*

*To avoid ambiguities, the statements will be reformulated as follows in the discussion: "The results are primarily valid for the study region in West Africa, while a wider generalization of the findings should be done with caution and after repeating similar evaluation studies in other places".*

4) The soil moisture, terrestrial water storage, and actual evaporation assessments were carried out without recalibrating the model and therefore the results for these variables are subject to substantial uncertainty. This is supported by the fact that MSWEP, which was used to force GLEAM, does not exhibit good actual evaporation scores. The model should be recalibrated for each variable.

*Response:*

*Performance of MSWEP: In our opinion, MSWEP has very good scores for modelled actual evaporation compared to GLEAM as it always exceeds a Pearson correlation coefficient (r) of 0.9 (Figure 8), with an average r=0.94 for the entire VRB (Appendix A3), and it has the highest spatial pattern score (Esp=0.26) among all the rainfall datasets (Appendix A3).*

*Model recalibration: We would like to emphasize here that the model is indeed recalibrated for each meteorological input product combination (i.e. rainfall and temperature), but it is recalibrated with streamflow (Q) only and not with soil moisture (Su), terrestrial water storage (St), and actual evaporation (Ea). The logic behind this approach is twofold:*
*i) We would like to know how well the model performs in combination with the different input variables; we therefore use Su, St and Ea as evaluation variables.*
*ii) Further calibrating our model with Su, St, and Ea would lead to additional model improvement due to the information content of these variables as demonstrated by Dembélé et al. (2020). In this case, it becomes difficult to disentangle the contribution of the rainfall datasets and the contribution of the calibration variables (Su, St, and Ea) to the overall model performance. Calibrating the model on one reference output variable (in-situ streamflow) and evaluating it against other output variables remains in our view a powerful method to assess the usefulness of a meteorological input dataset for hydrological modelling. By calibrating on streamflow, we give each meteorological data set "a chance" to perform as well as possible for streamflow; we then further discriminate between the usefulness of the input variables for hydrological modelling by assessing whether they can do a good job for streamflow and Su, St and Ea simultaneously. The "dream" input variable should indeed perform well for all variables if only calibrated on one.*

*We would like to emphasize here, that the evaluation of Su, St, and Ea is not done with the absolute values (i.e. raw data) of the satellite products, but rather we evaluate their temporal dynamics and spatial patterns using bias-insensitive metrics. Therefore, we substantially mitigate uncertainties that might arise from the assessment of these variables when using their absolute values (Dembélé et al. 2020; Nijzink et al., 2018; Mendiguren et al., 2017; Wambura et al., 2018). We had already discussed our choice for the Q-only calibration and its limitations at lines 456-458 and the potential uncertainties related to the satellite datasets used for evaluation at lines 441-443. However, we will now make the choice of the Q-only calibration clearer by adding the following in the discussion: "The model is calibrated only on Q data despite the known limitations of the Q-only calibration (Demirel et al., 2018). However, calibrating the model on additional variables would result in additional improvement in model performance that would not be separable from the contribution of the input datasets to the model performance. Therefore, regarding the goal of this study, the Q-only calibration was the best option to obtain the impact of various meteorological forcing datasets on the plausibility of hydrological processes."*

5) The word "gauge" is not used in the abstract and the datasets are only classified as either satellite or reanalysis. However, the amount of gauge data incorporated in the datasets may well be the overriding factor in determining the performance, given the good performance of TAMSAT and CHIRPS in terms of streamflow.

*Response: In the abstract, we will modify the statement "Seventeen precipitation products based on satellite data (…)" into "Seventeen precipitation products based essentially on gauge-corrected satellite data (…)". Moreover, Table 1 provide information on rainfall datasets developed with gauge data.*

6) Figures 7 and 10 are impossible to interpret, way too much information. Should be condensed.

**Response:** We agree with referee#1 that Figure 7 and 10 contain a lot of information. We will replace Figure 7 and 10 by Figure S30 and S48, respectively. Thereby, only showing the model performance for the entire VRB, while the performance for the four climatic zones will be moved to the supplementary materials.

[Figure]

Figure S30. Spatial pattern efficiency ($E_{SP}$) of soil moisture ($S_u$) over the entire simulation period (2003-2012) for the Volta River basin (VRB), using different combinations of precipitation and temperature products for

hydrological modelling. Each boxplot has 120 values corresponding to the number of months. The boxplots are colored from the best (green) to the worst performance (red) based on the median value.

[Figure]

Figure S48. Spatial pattern efficiency ($E_{SP}$) of actual evaporation ($E_a$) over the entire simulation period (2003-2012) for the Volta River basin (VRB), using different combinations of precipitation and temperature products for hydrological modelling. Each boxplot has 120 values corresponding to the number of months. The boxplots are colored from the best (green) to the worst performance (red) based on the median value.

**References**

Beck, H. E., Vergopolan, N., Pan, M., Levizzani, V., van Dijk, A. I. J. M., Weedon, G. P., Brocca, L., Pappenberger, F., Huffman, G. J., and Wood, E. F.: Global-scale evaluation of 22 precipitation datasets using gauge observations and hydrological modeling, Hydrol Earth Syst Sc, 21, 6201-6217, https://doi.org/10.5194/hess-21-6201-2017, 2017b.

Dembélé, M., Hrachowitz, M., Savenije, H. H., Mariéthoz, G., and Schaefli, B.: Improving the predictive skill of a distributed hydrological model by calibration on spatial patterns with multiple satellite datasets, Water Resources Research, https://doi.org/10.1029/2019WR026085, 2020.

Demirel, M. C., J. Mai, G. Mendiguren, J. Koch, L. Samaniego, and S. Stisen (2018), Combining satellite data and appropriate objective functions for improved spatial pattern performance of a distributed hydrologic model, Hydrology and Earth System Sciences, 22(2), 1299-1315, https://doi.org/10.5194/hess-22-1299-2018.

Mendiguren, G., J. Koch, and S. Stisen (2017), Spatial pattern evaluation of a calibrated national hydrological model - a remote-sensing-based diagnostic approach, Hydrology and Earth System Sciences, 21(12), 5987-6005, https://doi.org/10.5194/hess-21-5987-2017.

Nijzink, R. C., et al. (2018), Constraining Conceptual Hydrological Models With Multiple Information Sources, Water Resources Research, 54(10), 8332-8362, https://doi.org/10.1029/2017wr021895.

Wambura, F. J., O. Dietrich, and G. Lischeid (2018), Improving a distributed hydrological model using evapotranspiration-related boundary conditions as additional constraints in a data-scarce river basin, Hydrological processes, 32(6), 759-775, https://doi.org/10.1002/hyp.11453.

---

## Referee Comment (RC2) · Nadav Peleg (Referee) · 29 May 2020

Suitability of 17 rainfall and temperature gridded datasets for largescale hydrological modelling in West Africa

Dembélé et al. (2020)

**Review** (hess-2020-68)

In their paper, Dembélé et al. explore the suitability of combining time series of rainfall and temperature from different climate products as inputs into a hydrological model. The manuscript is well structured and written, methods are robust and results are presented adequately. The research question of the possibility of combining gridded climate variables from different sources to simulate various hydrological components is relevant and timely, and I believe will be of interest for the readers of HESS. Nevertheless, I have a few comments and suggestions for the authors to consider before I can recommend the paper for publication.

Sincerely,

Nadav Peleg

**Major comments**

1. I found one-step in the methodology (i.e. as presented in Figure 1) to be missing. I think it will be meaningful to know how the climate variables (rainfall, temperature) from each climate products are ranked in comparison to observed data (i.e. from ground stations) before ranking the 102 input combinations based on various hydrological components. I think this step is critical to understand the presented results. For example, JRA-55 and ERA-5 yield poor correlation with Ea (Figure 8), but isn't this because they are poorly reproducing the rainfall statistics over the VRB? GSMaP-std V6 reproduces well the streamflow (Figure 3), St (Figure 4), Su (Figure 5) and Ea (Figure 8) – will this product be ranked #1 when compared to ground stations? I assume there will be a high correlation between the ranks emerging from the comparison to ground stations and hydrological outputs from the model. If this case, wouldn't it be sufficient to evaluate the best products to use in hydrological simulations simply by comparing them to the few climate stations that are available in the catchment of interest or a nearby area? This is a point for discussion.

2. The modelling experiment includes 6 years for model calibration and 4 years for model evaluation. These are very short periods, not necessarily representing well the natural climatic and hydrological variability and not necessarily guarantying a successful calibration of the hydrological model parameters. First, I suggest demonstrating with a simple graph (can be presented as SI) that the natural variability is somehow represented in your 10-year data. Second, consider adding a short discussion regarding the sensitivity (quantified) of the hydrological model parameters to the short period that is used for the model training.

**Minor comments**

1. Usually, when considering using gridded climate variables from climate re-analysis/other products as inputs into hydrological models the following steps are taken: (i) computing the skills (i.e. temporal dynamics, magnitude, and occurrence) of the climate variables in comparison to observed data; (ii) choosing the (individual) climate product with the best skill to use; and (iii) performing a bias correction to the climate variables to improve the fit to the observed data. I am missing a paragraph in the introduction/discussion explaining why not simply following this practice which should improve the hydrological outputs from the model.

2. Results (Figure 3, for example). 22 values are used to represent the combined performance for the calibration and evaluation periods. This is not clear to me. Why not using a single Ekg value for the entire simulation period (merging the calibration and validation periods to a single period) for each gauge, i.e. 11 values in total per combination of temperature and precipitation product? What is the logic in separating the Ekg values to calibration and validation periods?

3. Table 1. I suggest adding in the table additional column indicating if the product refers to rainfall, temperature or both. Also, please double-check the space-time resolutions reporter in the table. I think that the CMORPH-CRT product, for example, has a resolution of 8-km and 30-min.

4. The use of second-order CV is interesting, I do not recall seeing it in the context of hydrological statistics. Why use it and not simply using Pearson's CV skill? A sentence explaining the motivation is needed.

5. Figures 7 and 10. Too many box-plots are presented. Perhaps present only the median (avoid using box-plots) to compare between products and climatic zones. This will considerably reduce the size and information plotted.

6. Generalization of the results. In lines 437-438 you mentioned that: "The results can be considered valid for West Africa and regions with similar hydroclimatic and physical features. A wider generalization of the findings should be done with caution and after repeating similar evaluation studies in other places". I do not think that you can generalize the results - they are likely to differ between locations as the quality of climate variables from different climate products differ between locations. In my view, the key message of your paper is that for each large catchment you should consider multiple sources of climate data to find the climate variables combination that is suitable for your region. The VRB is simply a case study used to demonstrate this point.

---

## Author Comment (AC2) · 16 Jun 2020

**Authors' Reply to Referee #2 (Dr. Nadav Peleg)**

**"Suitability of 17 rainfall and temperature gridded datasets for largescale hydrological modelling in West Africa" by Dembélé et al.**

**Review** (hess-2020-68)

In their paper, Dembélé et al. explore the suitability of combining time series of rainfall and temperature from different climate products as inputs into a hydrological model. The manuscript is well structured and written, methods are robust and results are presented adequately. The research question of the possibility of combining gridded climate variables from different sources to simulate various hydrological components is relevant and timely, and I believe will be of interest for the readers of HESS. Nevertheless, I have a few comments and suggestions for the authors to consider before I can recommend the paper for publication.

Sincerely,

Nadav Peleg

*Response: We thank the referee#2 for the positive overall appreciation of our work. Below we provide answers to the referee's comments.*

**Major comments**

**1.** I found one-step in the methodology (i.e. as presented in Figure 1) to be missing. I think it will be meaningful to know how the climate variables (rainfall, temperature) from each climate products are ranked in comparison to observed data (i.e. from ground stations) before ranking the 102 input combinations based on various hydrological components. I think this step is critical to understand the presented results. For example, JRA-55 and ERA5 yield poor correlation with Ea (Figure 8), but isn't this because they are poorly reproducing the rainfall statistics over the VRB? GSMaP-std V6 reproduces well the streamflow (Figure 3), St (Figure 4), Su (Figure 5) and Ea (Figure 8) – will this product be ranked #1 when compared to ground stations? I assume there will be a high correlation between the ranks emerging from the comparison to ground stations and hydrological outputs from the model. If this case, wouldn't it be sufficient to evaluate the best products to use in hydrological simulations simply by comparing them to the few climate stations that are available in the catchment of interest or a nearby area? This is a point for discussion.

*Response:*

*Comparison with ground observations*

*We agree with the referee that knowing the performance of the meteorological datasets in comparison with ground measurement could be an interesting starting point. However, it is noteworthy that the Volta River basin (VRB) in West Africa is a data scarce region, not like other places in Europe and USA (e.g. Beck et al., 2019a) where a large amount of ground measurements is widely and freely accessible. The few datasets collected by local organizations in the VRB are not easily accessible due to the transboundary nature of the basin that is shared among six countries. It took us one year to obtain streamflow data, which was further subject to a thorough gap-filling and quality control of time series (Dembélé et al., 2019).*

*The VRB region has a low density of meteorological stations (see Figure 1 of Dembélé and Zwart 2016; and Figure 1 of Satgé et al., 2020). A thorough evaluation of satellite/reanalysis datasets with ground measurements in the VRB cannot be limited to a few stations because the basin is about 415,600 km$^2$ (ten times the size of Switzerland), with a unique and complex climate (see Section 2.4 Study Area), and a strong spatial variability of rainfall.*

*Even in case of ground measurement availability, the validity of point-to-pixel comparison is questionable (e.g. JRA-55 is 1.25°, and ERA5 is 0.25°) because the gauge measurement will hardly represent the spatial variability of rainfall in a pixel. Moreover, the rainfall datasets used in our study are essentially gauge-corrected data. Therefore, a robust ground evaluation would require independent ground measurements that are not used in the development of the rainfall datasets (Beck et al., 2019a), which is a luxury in West Africa.*

*Validity of ground evaluation for hydrological modelling*

*The skill of a product in reproducing well ground measurement under a point-to-pixel evaluation does not necessarily guarantee its high performance for hydrological modelling, mainly in complex hydroclimatic environments such as the VRB. The performance of isolated pixels might not be representative of all pixels. Usually, hydrological modelling is undertaken at daily or higher temporal resolution. However, mismatches between gauge and satellite reporting times are a major issue in ground evaluation (Beck et al., 2019a). This is confirmed by the substantial increase in the evaluation performance of rainfall datasets from daily to monthly time scale (Dembélé and Zwart (2016); see Figure 3 vs. Figure 8 of Satgé et al. (2020)).*

*We will add the following to our discussions: "When comparing the results of this study to the findings of Satgé et al. (2020) based on a point-to-pixel evaluation of gridded rainfall datasets in West Africa, it is noticeable that the ground evaluation might lead to different results as compared to the hydrological evaluation as adopted in the current study. The skill of a rainfall product in reproducing well ground measurement under a point-to-pixel evaluation does not necessarily correlate with its performance for hydrological modelling, mainly in large and complex hydroclimatic environments such as the VRB. Therefore, ground evaluation it is not always a needful step before hydrological evaluation of gridded rainfall datasets."*

**2.** The modelling experiment includes 6 years for model calibration and 4 years for model evaluation. These are very short periods, not necessarily representing well the natural climatic and hydrological variability and not necessarily guarantying a successful calibration of the hydrological model parameters. First, I suggest demonstrating with a simple graph (can be presented as SI) that the natural variability is somehow represented in your 10-year data. Second, consider adding a short discussion regarding the sensitivity (quantified) of the hydrological model parameters to the short period that is used for the model training.

*Response:*

*Length of the calibration and simulation period*

*We agree that the modelling period of 10 years, which includes 6 years for calibration and 4 years for evaluation, might not seem very long, but is long enough to obtain a well calibrated model in our case, as previously demonstrated by Dembélé et al., (2020). Moreover, a 3-year model warm up period (2000-*

*2002) precedes the calibration period. The choice of the modelling period is constrained by the availability and the quality of the in-situ streamflow measurement in the data-scarce VRB (Dembélé et al., 2019).*

*Moreover, it is important to stress that we adopt a daily streamflow calibration, which means a time series of 2192 time steps to simulate and match for each of the 11 gauging stations during the 6-year model calibration period (2003-2008), or 3653 time steps for the 10-year simulation period (2003-2012). In our opinion, this is a robust model calibration approach, additionally supported by the fact that we adopt a multi-site calibration simultaneously at 11 streamflow gauging points located in very distinct hydroclimatic zones within the basin (see Figure 2). It is worth mentioning that the computational cost for each of the 102 input data combinations is about 6 days for 4000 parameter iterations during the model calibration on a computer Intel Xeon Processor E5-2697 v3 with 64 GB of RAM.*

*Natural variability of streamflow*

*We thank the referee#2 for this important comment on natural variability that was not appropriately discussed. Natural variability of daily streamflow can be observed at each of the 11 streamflow gauging sites used in this study, and inter-site variability of streamflow can be observed as well for the 10-year period (2003-2012). As it can be seen in Figure R1 below, the modelling period covers years with considerably different streamflow volumes during the wet season and with considerably different peak discharges, ranging e.g. for station 4 from 250 m3/s (year 2011) to 900 m3/s (year 2003). In general, years 2004, 2005 and 2009 can be considered as dry while 2003, 2006, and 2010 are wet for station 2 and 4, which have low flows as compared to the station 11.*

*Similar figures showing the hydrographs of all the eleven stations will be added to the supplementary materials and indexed in the text of the revised manuscript.*

[Figure]

*Figure R1: Hydrographs at three different gauging stations in the VRB during the modelling period comprised of the calibration period (2003-2008) and the evaluation period (2009-2012)*

*Natural variability can also be observed in the rainfall datasets as shown below in Figures R2-R3. It can be seen that rainfall varies both in time and space across different climatic zones in the VRB, which makes it an interesting case study for rainfall evaluation.*

*These figures will be added to the supplementary materials and indexed in the text of the revised manuscript.*

[Figure]

*Figure R2: Annual total rainfall for 17 datasets for different climatic zones in the VRB*

[Figure]

*Figure R3: Monthly total rainfall for 17 datasets for different climatic zones in the VRB*

*Sensitivity of model parameters*

*In the supplementary materials (Figure S65 at Section 9.2 on P61), we provide a figure that shows the distribution of each of the 36 global model parameters and their sensitivity (i.e. second-order coefficient of variation) to different input meteorological data. It can be seen that most of the model parameters vary considerably as a response to the change of rainfall and temperature data.*

*The following will be added to the discussion: "(…). Moreover, it can be noticed that most of the model parameters are sensitive to the change in meteorological input datasets (Figure S65)."*

**Minor comments**

**1.** Usually, when considering using gridded climate variables from climate re-analysis/other products as inputs into hydrological models the following steps are taken: (i) computing the skills (i.e. temporal dynamics, magnitude, and occurrence) of the climate variables in comparison to observed data; (ii) choosing the (individual) climate product with the best skill to use; and (iii) performing a bias correction to the climate variables to improve the fit to the observed data. I am missing a paragraph in the introduction/discussion explaining why not simply following this practice which should improve the hydrological outputs from the model.

*Response: The approach proposed by the referee#2 is usually applied for hydrological climate change impact studies, where climate projection data known to be biased are first evaluated and corrected with observations. In our manuscript, we have described at lines 78-85 the usually adopted approaches for the evaluation of gridded (satellite and reanalysis) datasets as follows:*
*"The errors quantification of SRPs and reanalysis products is usually done by comparing them with in-situ measurements (e.g. Dembélé and Zwart, 2016;Thiemig et al., 2012;Beck et al., 2019a;Caroletti et al., 2019;Satgé et al., 2020), or by assessing their reliability as forcing for hydrological models (e.g.Duethmann et al., 2013;Pan et al., 2010;Nkiaka et al., 2017). Other evaluation approaches include triple collocation, which is a technique that estimates the variance of unknown errors of three independent variables without a reference or observed variable (e.g. Massari et al., 2017;Alemohammad et al., 2015;McColl et al., 2014;Roebeling et al., 2012). Compared to the ground-truthing approach, the hydrological evaluation approach has received limited attention (Camici et al., 2018;Poméon et al., 2017)."*

*Among those approaches, we adopted the hydrological evaluation, which consists in assessing the reliability of the gridded datasets in reproducing plausible spatiotemporal patterns of hydrological processes when used as input to a model, knowing that they might still present some discrepancies with ground measurements. This approach is particularly interesting in data scarce regions where ground evaluation is challenging or impossible. It is important to mention here that the gridded datasets that we are evaluating in our study are essentially gauge-corrected datasets as mentioned at lines 160-161, also see Table 1. In this case, the datasets are already bias-corrected.*

*We will make this clearer in the abstract by mentioning the use of gauge-corrected datasets in our study. Therefore, as also requested by the referee#1, the statement "Seventeen precipitation products based on satellite data (…)" will be replaced by "Seventeen precipitation products based essentially on gauge-corrected satellite data (…)". Moreover, Table 1 provides information on rainfall datasets developed with gauge data.*

**2.** Results (Figure 3, for example). 22 values are used to represent the combined performance for the calibration and evaluation periods. This is not clear to me. Why not using a single Ekg value for the entire simulation period (merging the calibration and validation periods to a single period) for each gauge, i.e. 11 values in total per combination of temperature and precipitation product? What is the logic in separating the Ekg values to calibration and validation periods?

*Response: The decision for using 22 values of $E_{KG}$ (11 for calibration + 11 for evaluation) was based on the necessity to have enough elements for plotting the boxplots. For simplicity in reporting, a new Figure 3 will be provided only showing the median $E_{KG}$ of the entire simulation period, similarly to Figure 4, 5 and 8.*

**3.** Table 1. I suggest adding in the table additional column indicating if the product refers to rainfall, temperature or both. Also, please double-check the space-time resolutions reporter in the table. I think that the CMORPH-CRT product, for example, has a resolution of 8-km and 30-min.

*Response: We thank the referee for the suggestion, which will be considered in the revised manuscript. We are aware that different versions of the datasets exist, so that we have carefully mentioned in the caption of Table 1 that the information provided refer to the version of the datasets we have used. The provided information for CMORPH-CRT is correct, and the data was accessed from this web link: [ftp://ftp.cpc.ncep.noaa.gov/precip/CMORPH_V1.0/CRT/0.25deg-DLY_00Z/](ftp://ftp.cpc.ncep.noaa.gov/precip/CMORPH_V1.0/CRT/0.25deg-DLY_00Z/)*

**4.** The use of second-order CV is interesting, I do not recall seeing it in the context of hydrological statistics. Why use it and not simply using Pearson's CV skill? A sentence explaining the motivation is needed.

*Response: We agree with the referee that the use of the second-order CV is uncommon. In the revised manuscript, we will add the reasons of it use instead of the classical Pearson's CV, which has major limitations that are comprehensively described by Kvålseth (2017).*

**5.** Figures 7 and 10. Too many box-plots are presented. Perhaps present only the median (avoid using box-plots) to compare between products and climatic zones. This will considerably reduce the size and information plotted.

*Response: We agree with the referee#2 that Figure 7 and 10 contain a lot of information. As also responded to the referee#1 (comment #6), we will replace Figure 7 and 10 by Figure S30 and S48, respectively. Thereby, only showing the model performance for the entire VRB, while the performance for the four climatic zones will be moved to the supplementary materials.*

**6.** Generalization of the results. In lines 437-438 you mentioned that: "The results can be considered valid for West Africa and regions with similar hydroclimatic and physical features. A wider generalization of the findings should be done with caution and after repeating similar evaluation studies in other places". I do not think that you can generalize the results - they are likely to differ between locations as the quality of climate variables from different climate products differ between locations. In my view, the key message of your paper is that for each large catchment you should consider multiple sources of climate data to find the climate variables combination that is suitable for your region. The VRB is simply a case study used to demonstrate this point.

*__Response__: We agree with the referee#2 and we would like to stress that we did not intend to generalize our results as we carefully draw the reader's attention on the necessity to repeat the same experiment in other regions. As also responded to the referee#1 (comment #3), and to avoid ambiguities, the statement will be modified as follows: "The results are primarily valid for the study region in West Africa, while a wider generalization of the findings should be done with caution and after repeating similar evaluation studies in other places".*

**References**

Beck, H. E., Pan, M., Roy, T., Weedon, G. P., Pappenberger, F., Van Dijk, A. I., Huffman, G. J., Adler, R. F., and Wood, E. F.: Daily evaluation of 26 precipitation datasets using Stage-IV gauge-radar data for the CONUS, Hydrol Earth Syst Sc, 23, 207-224, https://doi.org/10.5194/hess-23-207-2019, 2019a.

Dembélé, M., and Zwart, S. J.: Evaluation and comparison of satellite-based rainfall products in Burkina Faso, West Africa, Int J Remote Sens, 37, 3995-4014, https://doi.org/10.1080/01431161.2016.1207258, 2016.

Dembélé, M., Oriani, F., Tumbulto, J., Mariéthoz, G., and Schaefli, B.: Gap-filling of daily streamflow time series using Direct Sampling in various hydroclimatic settings, Journal of Hydrology, 569, 573-586, https://doi.org/10.1016/j.jhydrol.2018.11.076, 2019.

Dembélé, M., Hrachowitz, M., Savenije, H. H., Mariéthoz, G., and Schaefli, B.: Improving the predictive skill of a distributed hydrological model by calibration on spatial patterns with multiple satellite datasets, Water Resources Research, e2019WR026085, https://doi.org/10.1029/2019WR026085, 2020a.

Kvålseth, T. O.: Coefficient of variation: the second-order alternative, Journal of Applied Statistics, 44, 402-415, https://doi.org/10.1080/02664763.2016.1174195, 2017.

Nash, J. E., and J. V. Sutcliffe (1970), River flow forecasting through conceptual models part I—A discussion of principles, Journal of hydrology, 10(3), 282-290, https://doi.org/10.1016/0022-1694(70)90255-6.

Satgé, F., Defrance, D., Sultan, B., Bonnet, M.-P., Seyler, F., Rouché, N., Pierron, F., and Paturel, J.-E.: Evaluation of 23 gridded precipitation datasets across West Africa, Journal of Hydrology, 581, 124412, https://doi.org/10.1016/j.jhydrol.2019.124412, 2020.

---

## Author Comment (AC3) · 19 Jun 2020

**Corrigendum for**

**"Suitability of 17 rainfall and temperature gridded datasets for largescale hydrological modelling in West Africa" by Dembélé et al.**

**Author's report**

In our paper HESS-2020-68 submitted to HESSD, we have realized an erroneous reporting of the objective function used for model calibration with streamflow data. Instead of using $E_{KG}$ as currently reported in the manuscript, we used a combination of the Nash-Sutcliffe efficiency (Nash and Sutcliffe, 1970) of streamflow ($E_{NS}$) and the Nash-Sutcliffe efficiency of the logarithm of streamflow ($E_{NSlog}$), similarly to Dembélé et al. (2020). This setting has the advantage of identifying a parameter set that better predicts both high and low flows because $E_{NS}$ is known to be very sensitive to high flows, while $E_{NSlog}$ is a metric for low flows (Krause et al., 2005; Oudin et al., 2006; Pushpalatha et al., 2012).

Therefore, the current objective function (Eq.3) will be replaced by the following in the revised manuscript:

$$\Phi_Q = \frac{1}{g}\sum_1^g \sqrt{(1 - E_{NS})^2 + \left(1 - E_{NSlog}\right)^2}, \text{ with} \tag{3}$$

$$E_{NS} = 1 - \frac{\sum_1^t\left(Q_{mod}(t) - Q_{obs}(t)\right)^2}{\sum_1^t(Q_{obs}(t) - \overline{Q_{obs}})^2} \text{ and} \tag{4}$$

$$E_{NSlog} = 1 - \frac{\sum_1^t\left[\log(Q_{mod}(t)) - \log(Q_{obs}(t))\right]^2}{\sum_1^t\left[\log(Q_{obs}(t)) - \overline{\log(Q_{obs})}\right]^2} \tag{5}$$

where $Q_{mod}$ and $Q_{obs}$ are the modelled and observed streamflow, $t$ is the number of time steps of the calibration period, and $g$ is the number of streamflow gauging stations present within the modelling domain.

These modifications do not affect the current results, rather they reinforce the analysis as we will report on the model performance for streamflow with multiple skill scores (i.e. $E_{NS}$, $E_{NSlog}$ and $E_{KG}$). Consequently, we will additionally report on the model performance for streamflow using $E_{NS}$ and $E_{NSlog}$ in the revised manuscript. Changes will be made to Appendix A3, and Figure 3 will be modified as follows:

[Figure]

Figure 3. Median Kling-Gupta efficiency ($E_{KG}$), Nash-Sutcliffe efficiency ($E_{NS}$) and Nash-Sutcliffe efficiency of the logarithm ($E_{NSlog}$) of daily streamflow ($Q$) over the simulation period (2003-2012) for 102 combinations of 17 rainfall datasets (y-axis) and 6 temperature datasets (x-axis) used as forcing for the hydrological model.

Additional figures with detailed statistics and boxplots will be added to the supplementary materials.

**References**

Dembélé, M., Hrachowitz, M., Savenije, H. H., Mariéthoz, G., and Schaefli, B.: Improving the predictive skill of a distributed hydrological model by calibration on spatial patterns with multiple satellite datasets, Water Resources Research, e2019WR026085, https://doi.org/10.1029/2019WR026085, 2020a.

Krause, P., D. Boyle, and F. Bäse (2005), Comparison of different efficiency criteria for hydrological model assessment, Advances in geosciences, 5, 89-97, https://doi.org/10.5194/adgeo-5-89-2005.

Nash, J. E., and J. V. Sutcliffe (1970), River flow forecasting through conceptual models part I—A discussion of principles, Journal of hydrology, 10(3), 282-290, https://doi.org/10.1016/0022-1694(70)90255-6.

Oudin, L., V. Andréassian, T. Mathevet, C. Perrin, and C. Michel (2006), Dynamic averaging of rainfall-runoff model simulations from complementary model parameterizations, Water Resources Research, 42(7), https://doi.org/10.1029/2005WR004636.

Pushpalatha, R., C. Perrin, N. Le Moine, and V. Andreassian (2012), A review of efficiency criteria suitable for evaluating low-flow simulations, Journal of Hydrology, 420, 171-182, https://doi.org/10.1016/j.jhydrol.2011.11.055.

---

## Author Response (AR2)

**Authors' response to the reviews of**

**"Suitability of 17 rainfall and temperature gridded datasets for largescale hydrological modelling in West Africa" by Dembélé et al. (HESS-2020-68)**

**Reply to the Editor**

Dear authors

Please address all comments by the reviewer. I thnk the reviewer has a point with the comment regarding the scope .I look forward to the revised manuscript.

Sincerely,

Albrecht Weerts

*Response:*

*Dear Prof. Albrecht Weerts,*

*We thank you for handling our manuscript. Below we provide point-by-point responses to the referees' comments and indicate modifications done to our manuscript.*

*Best regards,*

*Moctar Dembélé*
*on behalf of all co-authors*

**Reply to Referee #2 – Dr Nadav Peleg**

Dear Moctar Dembélé and co-authors,

Thank you for your detailed replies to my comments from the first round. Reading the revised manuscript, I found it improved in comparison to the original submission. There is one point that I feel can be still improved. Reading of the title and abstract, the paper still reads like a case-study specific to West Africa region. In your replies, you discussed the point that the need to investigate the suitability of different climatic dataset to the hydrology comes from the fact that you cannot compare the datasets to climatic ground stations due to their (lack of) availability. You do mention "data-scarce regions" as one of your keywords, but do not refer to this point in your paper. Why not emphasizing the lack of climatic data as your motivation to explore different combinations of climatic datasets? That will make your paper a bit less "case-study" oriented and more appealing to the general readers. Several small modifications will be required: changing the title from "Suitability of 17 rainfall and temperature gridded datasets for largescale hydrological modelling in West Africa" to something like "Suitability of 17 rainfall and temperature gridded datasets for largescale hydrological modelling in data-scarce regions", for example. Of course, edits will be required in the abstract (adding one sentence likely) and a paragraph will need to be added in the introduction and discussion parts. This is just a suggestion - the paper can be accepted as is, but I really think that modifying the storyline a bit can make it more appealing.

*__Response__: We thank reviewer#2 (Nadav Peleg) for his thorough evaluation of our manuscript and for his positive appreciation of our work.*

*We appreciate the suggestions of the reviewer and we have taken the following actions:*

*-The title is kept as is because although our study focuses on a data-scarce region, we only study the West African region. This will avoid raising too much the expectation of some readers who might be interested in other regions than West Africa.*

*-The last sentence of the abstract is modified to motivate the importance of our work for data-scarce regions as follows: "Finally, some region-tailored datasets outperform the global datasets, thereby stressing the necessity and importance of regional evaluation studies for satellite and reanalysis meteorological datasets, which are increasingly becoming an alternative to in situ measurements in data scarce regions."*

*-The introduction and discussions already mention "data-scarce" issues at lines 57, 122, and the necessity of hydrological evaluation as compared to ground evaluation at lines 465-4694. However, a paragraph is added to section 2.4 to discuss the issue of data availability and the limitations of ground measurements, which motivate the hydrological evaluation of the rainfall datasets (see our response to your minor comment #1).*

Some other minor comments:

1. In section 2.4: I suggest adding information regarding data availability (stream gauges and climate stations) even if you decide not to change the storyline. Observed data is only reported in Figure 2 and is not detailed.

*Done. The following is added to section 2.4:*

*"The VRB is a data-scarce region, not like other places in Europe and USA where a large amount of ground measurements is widely and freely accessible. The few datasets collected by local organizations in the VRB are not easily accessible due to the transboundary nature of the basin that is shared among six countries. Moreover, the VRB region has a low density of meteorological stations (cf. Figure 1 of Dembélé and Zwart 2016; and Figure 1 of Satgé et al., 2020). A thorough evaluation of satellite and reanalysis datasets with ground measurements in the VRB cannot be limited to a few stations because of the large size of the basin and the strong spatial variability of rainfall. Moreover, a robust ground evaluation would require independent in situ measurements that are not used in the development of the SRPs and reanalysis datasets (Beck et al., 2019a), which is a luxury in West Africa. These limitations in in situ data availability further motivate the hydrological evaluation of SRPs and reanalysis datasets".*

2. Figure 1. What do you mean by morphological datasets? And - why is it in plural? Was more than one "morphological" dataset used?

*By morphological datasets we mean datasets that described the physical characteristics of the basin (e.g. topography, soil, geology, land cover) as described in Table 2. This is now made clearer by specifying "Morphological datasets (DEM, soil, geology, etc.)" in Figure 1.*

3. Figure 2. Consider replacing the blue colour of the Guinean zone with another colour as it is too similar to the colour of the river and lakes. In the figure caption, some words are required to describe panels (a) and (b). The legend of the elevation - I suggest removing the labels "High" and "Low", and adding a label for the 500 m elevation.

*Done.*

4. Table 3. The information presented here can be placed in the caption of Fig. 2 and the Table can be deleted.
*Done.*

Sincerely,
Nadav Peleg

*Thanks very much for your evaluation of our manuscript.*

*Moctar Dembélé*
*on behalf of all co-authors*

[revised manuscript text omitted]